# Embryonic origin of adult stem cells required for tissue homeostasis and regeneration

**Erin L Davies, Kai Lei, Christopher W Seidel, Amanda E Kroesen, Sean A McKinney, Longhua Guo, Sofia MC Robb, Eric J Ross, Kirsten Gotting, Alejandro Sánchez Alvarado\***

Howard Hughes Medical Institute, Stowers Institute for Medical Research, Kansas City, United States

**Abstract** Planarian neoblasts are pluripotent, adult somatic stem cells and lineage-primed progenitors that are required for the production and maintenance of all differentiated cell types, including the germline. Neoblasts, originally defined as undifferentiated cells residing in the adult parenchyma, are frequently compared to embryonic stem cells yet their developmental origin remains obscure. We investigated the provenance of neoblasts during *Schmidtea mediterranea* embryogenesis, and report that neoblasts arise from an anarchic, cycling *piwi-1+* population wholly responsible for production of all temporary and definitive organs during embryogenesis. Early embryonic *piwi-1+* cells are molecularly and functionally distinct from neoblasts: they express unique cohorts of early embryo enriched transcripts and behave differently than neoblasts in cell transplantation assays. Neoblast lineages arise as organogenesis begins and are required for construction of all major organ systems during embryogenesis. These subpopulations are continuously generated during adulthood, where they act as agents of tissue homeostasis and regeneration.

\*For correspondence: asa@ stowers.org

## Introduction

Neoblasts are planarian adult somatic stem cells that exhibit levels of plasticity and pluripotency comparable to embryonic and induced pluripotent stem cells (*Elliott and Sánchez Alvarado, 2013*; *Rink, 2013*; *Wagner et al., 2011*). In flies, fish, mice and humans, adult somatic stem cells are fate-restricted, sustaining production of cell lineage(s) in resident tissues (*Fuchs and Segre, 2000*; *Wagers and Weissman, 2004*). Although embryonic stem cells cultured ex vivo remain capable of producing a diversity of tissue types from different germ layers, such plasticity is typically lost from most somatic cells as development proceeds. In contrast, the planarian neoblast population is wholly responsible for the production of all differentiated cell types in these bilaterally symmetric, triplo-blastic animals (*Baguñà and Auladell, 1989*). In fact, transplantation of a single neoblast into a stem cell deficient host was sufficient for rescue and long-term reconstitution (*Wagner et al., 2011*), confirming the pluripotency of planarian somatic stem cells. Despite longstanding discussion of the similarities between neoblasts and embryonic stem cells, a comparison explicitly stated in the original definition of the term (*Randolph, 1892*), the provenance of neoblasts during embryogenesis was unknown.

Neoblasts are abundant and widely distributed across the anteroposterior axis, occupying the parenchymal space surrounding the gut (*Reddien et al., 2005*). All neoblasts contain chromatoid bodies (*Auladell et al., 1993*; *Hay and Coward, 1975*; *Hori, 1982*; *Morita et al., 1969*) and express nuage genes, including *piwi-1*, and factors implicated in germ cell identity, genome surveillance and

**eLife digest** Flatworms are masters of regeneration. If virtually any piece of a flatworm is cut off, a new fully functional individual will grow from it within two weeks. This is no simple task since flatworms contain a wide variety of organ systems, including a brain, nervous system, eyes, kidneys, gut, muscle and skin.

Flatworms owe their regenerative abilities to adult stem cells called neoblasts. Like embryonic stem cells, neoblasts can replicate themselves and they can develop into any type of cell found in an adult worm. In contrast, adult stem cells in fruit flies, zebrafish, mice and humans can only produce the type of cells found in the organ or tissue they live in.

Now, Davies et al. have tracked how and when neoblasts develop in embryos of the flatworm species *Schmidtea mediterranea* by documenting the distinct gene expression signatures in flatworm embryos at various stages of development. An atlas of the genes that are expressed in various embryonic tissues and in major organs as they begin to develop was also created. These tools, and the results of cell transplantation experiments, revealed that neoblasts emerge from embryonic stem cells as the major organs start to form. As the emerging neoblasts start to express the same combination of genes as adult neoblasts, they also begin to behave just like these cells. The populations of neoblasts remain present throughout the life of the flatworm, helping to maintain, repair and regenerate tissues.

In the future, work that builds on the results presented here by Davies et al. will help researchers to understand more about how stem cells are maintained and regulated. By learning more about the genetic differences between neoblasts and human adult stem cells scientists may be able to explain why humans and other mammals have a limited ability to regenerate. This information could potentially help to develop treatments that stimulate regeneration in patients with degenerative diseases or traumatic injuries.

post-transcriptional regulation of gene expression (*Guo et al., 2006*; *Palakodeti et al., 2008*; *Reddien et al., 2005*; *Rouhana et al., 2010*, *2012*; *Salvetti et al., 2005*; *Shibata et al., 1999*; *Solana et al., 2009*; *Wagner et al., 2012*; *Yoshida-Kashikawa et al., 2007*). Neoblasts are the only cycling somatic cells in adults (*Baguñà, 1976*; *Newmark and Sánchez Alvarado, 2000*; *Orii et al., 2005*; *Salvetti et al., 2000*); quiescent neoblasts were not observed in BrdU pulse chase experiments (*Newmark and Sánchez Alvarado, 2000*). Mounting evidence suggests that the neoblast population contains pluripotent stem cells as well as cycling, lineage-primed progenitors (*Reddien, 2013*). Heterogeneous expression of developmental transcription factors (TFs) in neoblasts has been reported and likely reflects the diversity of lineage-primed progenitors within the compartment (*Adler et al., 2014*; *Cowles et al., 2013*; *Currie and Pearson, 2013*; *Lapan and Reddien, 2011*, *2012*; *März et al., 2013*; *Pearson and Sánchez Alvarado, 2010*; *Scimone et al., 2014*, *2011*; *van Wolfswinkel et al., 2014*; *Wenemoser et al., 2012*).

*Schmidtea mediterranea* (*Smed*) freshwater flatworms are stable diploids that exist as two biotypes: asexual animals that reproduce by fission, and obligate cross-fertilizing hermaphrodites that reproduce sexually (*Newmark and Sánchez Alvarado, 2002*; *Newmark et al., 2008*). Both biotypes mount robust regeneration responses following amputation, and similarly rely on neoblasts for homeostatic maintenance and regeneration of all tissues. The asexual clonal line CIW4 (C4) has received the most scrutiny in studies examining the molecular mechanisms underlying regeneration, neoblast maintenance, pluripotency and lineage commitment (*Newmark and Sánchez Alvarado, 2002*). However, neoblasts are ever-present in C4 animals, precluding investigation of their developmental origin. Neither a normal table for *Smed* embryonic development nor functional studies have been reported.

Our work establishes *Smed* as a developmental model system and leverages the novel, unexploited context of embryogenesis to hone the molecular and operational definition of the planarian neoblast. We generated a molecular staging resource for *Smed* embryogenesis that associates unique gene expression signatures with chronological age, embryo morphology, representative images and written summaries of key developmental events to holistically describe and define

prototypes for each stage. We also provide an atlas of molecular markers describing temporary embryonic tissue types and definitive organ system development. These data, found in the supplementary material, are also searchable online at https://planosphere.stowers.org.

We investigated the developmental origin of neoblasts during *Smed* embryogenesis and show that early embryonic cells are molecularly and functionally distinct from the adult neoblast population. Pluripotent neoblasts and lineage-dedicated progenitors arise as organogenesis begins. Our results suggest that the framework for understanding cell fate specification and organ formation during *Smed* embryogenesis diverges radically from existing developmental paradigms. Here, in a bilaterally symmetric, triploblastic animal not thought to undergo gastrulation (*Cardona et al., 2005*; *Le Moigne, 1963*; *Sánchez Alvarado, 2003*; *Stevens, 1904*), heterogeneous expression of key developmental regulators within a pluripotent, cycling blastomere population generates the panoply of lineage-dedicated progenitors required for organogenesis. Moreover, neoblasts perpetuate embryonic developmental programs during adulthood, where they are required for continued maintenance and rebuilding of tissues during homeostasis and regeneration.

## Results

### A molecular staging series for *Smed* embryogenesis informed by single embryo RNA-Seq

*Smed* flatworms are direct developers: newborn hatchlings grow and mature into adult worms without an intervening larval stage (*Sánchez Alvarado, 2003*). At hatching, juveniles are sexually immature but otherwise possess a body plan grossly similar to that of adult hermaphrodites (*Sánchez Alvarado, 2003*; *Wang et al., 2007*). *Smed* embryos undergo an evolutionarily divergent mode of development that bears little resemblance to the ancestral Spiralian cleavage programs utilized by many Lophotrochozoans. *Smed* embryos are ectolecithal: yolk is not contained within oocytes, but rather is produced by somatic vitellaria (yolk glands) arrayed ventrolaterally beneath the testes (*Chong et al., 2011*; *Steiner et al., 2016*; *Stevens, 1904*). Oocytes are fertilized internally by sperm from a partner. Zygote(s) are packaged, along with yolk cells, into an egg capsule in the genital atrium (*Figure 1A*) (*Chong et al., 2011*; *Hyman, 1951*; *Newmark et al., 2008*; *Stevens, 1904*).

In contrast to the synchronous, oriented blastomere cleavage patterns of Spiralian embryos (*Lambert, 2010*), blastomeres in freshwater planarian embryos undergo dispersed cleavage among yolk cells: they divide asynchronously and are not in direct contact with one another (*Bardeen, 1902*; *Cardona et al., 2005*; *Hallez, 1887*; *Ijima, 1884*; *Le Moigne, 1963*; *Metschnikoff, 1883*; *Vara et al., 2008*). During sphere formation, some blastomeres differentiate into temporary embryonic cell types that provide form and function to the embryo, including the primitive ectoderm, temporary embryonic pharynx and primitive gut (*Cardona et al., 2005*; *Hallez, 1887*; *Le Moigne, 1963*; *Sánchez Alvarado, 2003*). Temporary embryonic tissues are not thought to contribute to the juvenile body plan; they are thought to degenerate as the definitive organs form and morphogenesis proceeds (*Cardona et al., 2005*; *Le Moigne, 1963*; *Vara et al., 2008*). In contrast, undifferentiated blastomeres remaining after sphere formation are thought to give rise to all definitive tissues found in juvenile worms (*Hallez, 1887*; *Hyman, 1951*; *Le Moigne, 1963*; *Sánchez Alvarado, 2003*; *Stevens, 1904*).

Several attributes of ectolecithal development challenge efforts to accurately stage live embryos including internal fertilization, dispersed cleavage, lack of morphological landmarks, and inherent variability in embryo size and developmental timing. Extant staging series for *Polycelis nigra* (*Le Moigne, 1963*), *Schmidtea polychroa* (*Cardona et al., 2005*; *Martín-Durán et al., 2010*) and *Girardia tigrina* (*Vara et al., 2008*) rely upon gross morphological criteria gleaned from live animals and fixed, histologically or antibody-stained specimens. As this is the first systematic characterization of *Smed* embryogenesis, we established an accurate and objective staging method based on unique gene expression signatures, cohorts of enriched transcripts identified through single embryo RNA-Seq , associated with chronological age and embryo morphology (*Figure 1B–I*, *Figure 1—source data 1*). When appropriate, efforts were made to integrate and adapt extant staging schema from other planarian species.

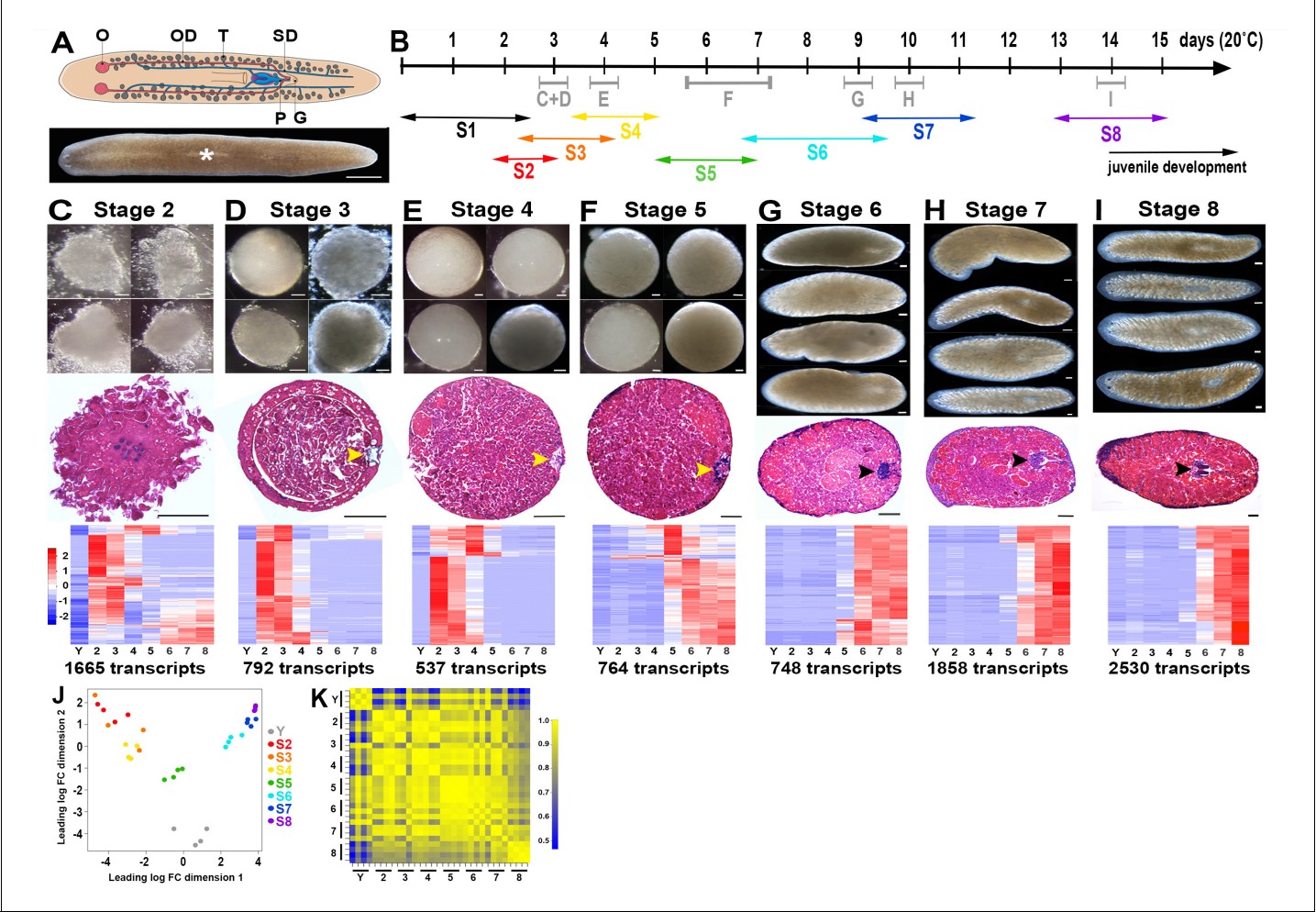

**Figure 1.** A molecular staging series for *Smed* embryogenesis informed by single embryo RNA-Seq. (**A**) Top: Cartoon depicting the reproductive system of a sexually mature *Smed* hermaphrodite. Ventral view. G, gonopore; O, ovary; OD, oviduct; P, penis papilla; SD, sperm duct . Oocytes are fertilized internally and zygote(s) are packaged with yolk produced by vitellogenic gland cells into a developing egg capsule in the genital atrium (purple). Capsules are laid through the gonopore. Bottom: Brightfield image of a live *Smed* hermaphrodite. Anterior: left. Dorsal view. White asterisk: pharynx. (**B**) Developmental timeline and staging designations for *Smed* embryogenesis at 20°C. Timeline: days (d) post egg capsule deposition. Gray bars and letters C–I indicate time windows, and corresponding panels (C–I), for RNA-Seq samples. Double-headed arrows: time windows for stages (S) S1–S8. (**C–I**) Brightfield images of live embryos harvested for RNA-Seq (top), hematoxylin- and eosin-stained sections (middle), and heat maps for enriched transcripts (bottom). Scale bars: 100 µm. Yellow arrowheads: temporary embryonic pharynx. Black arrowheads: definitive pharynx. Heat maps depict cohorts of enriched transcripts at indicated stages. (**J**) Principal component analysis demonstrates clustering of replicates and separation of developmental time points in expression space. (**K**) Correlation matrix for single embryo sequencing replicates. Total transcripts with a row sum >1 CPM: 31,248. (**C–K**) Y,yolk. 2, S2. 3, S3. 4, S4. 5, S5. 6, S6. 7, S7. 8, S8.

The following source data and figure supplements are available for figure 1:

**Source data 1.** Molecular staging resource for *Smed* embryogenesis.

**Source data 2.** Stage-2-enriched transcripts from pairwise and/or mixed stage reference comparisons.

**Source data 3.** Stage-3-enriched transcripts from pairwise and/or mixed stage reference comparisons.

**Source data 4.** Stage-4-enriched transcripts from pairwise and/or mixed stage reference comparisons.

**Source data 5.** Stage-5-enriched transcripts from pairwise and/or mixed stage reference comparisons.

**Source data 6.** Stage-6-enriched transcripts from pairwise and/or mixed stage reference comparisons.

*Figure 1 continued on next page*

*Figure 1 continued*

**Source data 7.** Stage-7-enriched transcripts from pairwise and/or mixed stage reference comparisons.

**Source data 8.** Stage-8-enriched transcripts from pairwise and/or mixed stage reference comparisons.

**Source data 9.** Molecular fate mapping resource.

**Figure supplement 1.** Histological cross-sections of Stage one embryos.

**Figure supplement 2.** MA plots for pairwise comparisons.

**Figure supplement 3.** MA plots for enriched transcripts identified in mixed-stage reference comparisons.

**Figure supplement 4.** Mean centered expression and average RPKM profiles for S2-enriched transcripts.

**Figure supplement 5.** Mean centered expression and average RPKM profiles for S3-enriched transcripts.

**Figure supplement 6.** Mean centered expression and average RPKM profiles for S4-enriched transcripts.

**Figure supplement 7.** Mean centered expression and average RPKM profiles for S5-enriched transcripts.

**Figure supplement 8.** Mean centered expression and average RPKM profiles for S6-enriched transcripts.

**Figure supplement 9.** Mean centered expression and average RPKM profiles for S7-enriched transcripts.

**Figure supplement 10.** Mean centered expression and average RPKM profiles for S8-enriched transcripts.

**Figure supplement 11.** Molecular markers for the primitive ectoderm.

**Figure supplement 12.** Molecular markers for the temporary embryonic pharynx.

**Figure supplement 13.** Molecular markers for the developing gut.

**Figure supplement 14.** :Molecular markers for the definitive pharynx.

**Figure supplement 15.** Molecular markers for the definitive epidermis.

**Figure supplement 16.** Molecular markers for the developing nervous system.

**Figure supplement 17.** Molecular markers for the developing musculature.

**Figure supplement 18.** Molecular markers for the developing excretory system.

**Figure supplement 19.** Molecular markers for the developing eyes.

*Smed* embryos gestate for approximately two weeks at 20°C prior to hatching. We generated total RNA replicates from single *Smed* embryos for seven chronologically and/or morphologically distinct stages (S), S2-–S8 (*Figure 1B–I*); S1 samples (zygotes, *Figure 1—figure supplement 1*) were not queried by RNA-SSeq. Yolk (Y) replicates were prepared from egg capsules lacking developing embryos at 8 days post capsule deposition. In addition, single animal replicates were prepared from C4 and virgin sexually mature adults (SX). RNA-Seq libraries were analyzed for four biological replicates (i.e., four individuals) per stage (Materials and methods). Identification of appropriate reference(s) and normalization methods was challenging due to vast differences in sample composition and complexity among different stages. However, clustering of replicates and discrete separation of stages was seen in a multidimensional scaling plot (*Figure 1J*). Replicates for a given stage generally

showed strong correlation among themselves despite not having controlled for differences in genetic background or embryo size (*Figure 1K*). Notably the S2 and S3 replicates, generated from embryos undergoing sphere formation and nascent spheres, respectively, were intermingled in expression space, showed the greatest variability and few significant differences in gene expression (*Figure 1J–K*, *Figure 1—source data 1*, *Figure 1—figure supplement 2*). Discrete stages were retained due to apparent differences in embryo morphology (*Figure 1C–D*). Similarities among S2 and S3 samples may be due to maternal RNA contribution or to difficulty detecting labile S2 embryos, such that only well-developed S2 embryos were prospected by RNA-Seq.

Two approaches were used to identify differentially expressed transcripts: pairwise comparisons of adjacent stages (*Figure 1—source data 1*, *Figure 1—figure supplement 2*)and comparisons of each stage relative to a mixed stage reference generated by averaging the read counts for the remaining replicates (Y, S2–S8) (Materials and methods, *Figure 1—source data 1*, *Figure 1—figure supplement 3*). The goal of the pairwise comparisons was to identify transcripts with the starkest changes in expression in either direction, without constraints on transcript behavior at other points during embryogenesis. In contrast, the mixed stage reference comparisons maximized the likelihood of identifying transcripts exhibiting stage-specific expression, and only upregulated transcripts were analyzed. Stringent criteria were applied in both scenarios for flagging differentially expressed transcripts, including thresholds based on the Benjamini-Hochberg adjusted p-value, fold-change, normalized RPKM expression level for time points, and identification of at least one open reading frame in the transcript (*Figure 1—source data 1*). More upregulated transcripts were identified in the mixed stage reference than in pairwise comparisons, perhaps due to increased sequencing depth of the averaged reference samples, which may enable more reliable detection of lowly expressed transcripts. Furthermore, identification of S2-enriched transcripts suggested that the whole embryo sequencing approach was sensitive enough to detect transcripts expressed in rare cell populations (i.e., blastomeres and differentiated cells making up the embryo proper). Non-redundant lists of upregulated transcripts, resulting from the union of the pairwise and mixed reference comparisons, served as molecular fingerprints for each time point for downstream analyses, including hierarchical clustering and GO analysis (*Figure 1—source data 1*). The molecular staging resource (*Figure 1—source data 1*) incorporates representative images (*Figure 1C–I*, *Figure 1—figure supplement 1*), gene expression signatures (*Figure 1—figure supplements 4–10*, *Figure 1—source data 2–8*) and written summaries of key developmental events (*Figure 1—source data 1*), defining S1–S8. An expression atlas describing temporary embryonic tissue types and development of the definitive organ systems is also provided (*Figure 1—source data 9*, *Figure 1—figure supplements 11–19*). The molecular staging resource and expression atlas are also available and searchable online (https://planosphere.stowers.org).

## Anarchic, cycling *piwi-1+* blastomeres fuel *Smed* embryonic development

The molecular staging series identified S2- and S3-enriched transcripts (*Figure 1C–D*, *Figure 1—source data 2–3*), including *elongation factor 1a-like-1* (*EF1a-like-1*), expressed in all known embryonic cell populations in nascent spherical embryos (*Figure 2A–B*). During S2, primitive ectoderm cells are the first to differentiate (*Hallez, 1887*; *Ijima, 1884*; *Le Moigne, 1963*; *Metschnikoff, 1883*). *EF1a-like-1* was expressed in primitive ectoderm cells (*Figure 2A–B*, *Video 1*), which flatten, elaborate numerous cytoplasmic processes, and form a single-cell layer bounding the sphere (*Figure 1—source data 9*, *Figure 1—figure supplement 11*). *EF1a-like-1* expression was also detected in the temporary embryonic pharynx (*Figure 2A–B*, *Video 1*) an innervated pump containing neurons, radial muscle and associated epithelial cells (*Figure 1—source data 9*, *Figure 1—figure supplement 12*), and in the primitive endoderm (*Figure 2A*), which consists of an inner gut cavity and phagocytic cells associated with the temporary embryonic pharynx (*Figure 1—source data 9*, *Figure 1—figure supplement 13*). A population of undifferentiated blastomeres and yolk cells remain in the embryonic wall, the parenchymal space between the primitive ectoderm and endoderm, in nascent S3 spheres (*Figure 1D*) (*Hyman, 1951*; *Sánchez Alvarado, 2003*). *EF1a-like-1* was expressed in undifferentiated blastomeres, but not yolk cells, in S3 embryos (*Figure 2A–B*, *Video 1*).

*piwi-1+* cells were present throughout embryogenesis. *piwi-1* expression was detected in oocytes (*Figure 2C*), suggesting that zygotes contain *piwi-1* mRNA. Zygote-derived blastomeres undergoing dispersed cleavage among yolk cells during S2 also expressed *piwi-1* (*Figure 2D*). Costaining with

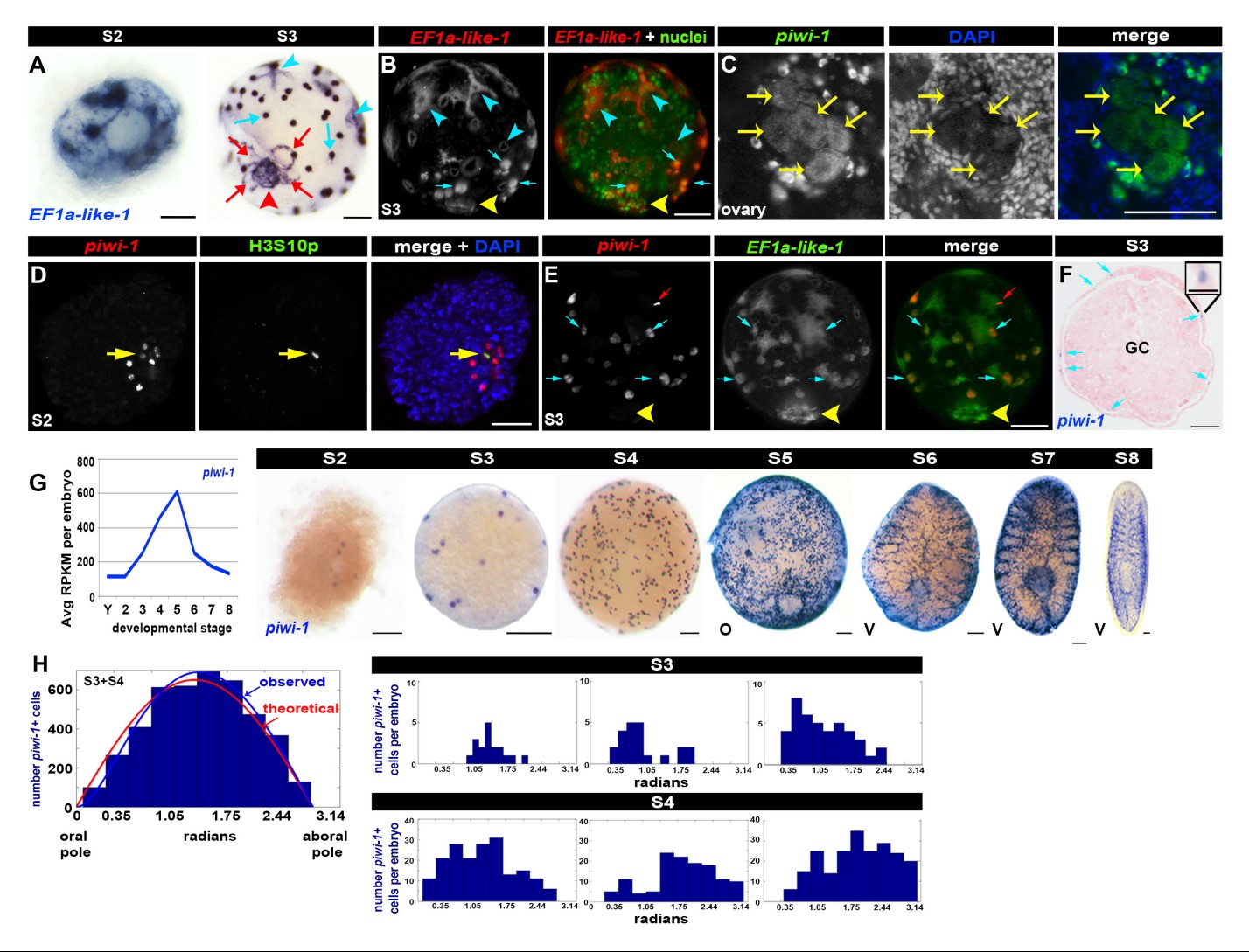

**Figure 2.** Blastomere anarchy drives *Smed* embryogenesis. (A–B) Architectural features of S2 and S3 embryos. (A) Expression of the pan embryonic cell marker *EF1a-like-1* (blue) in S2 (left) and S3 (right) embryos. Cyan arrowheads: primitive ectoderm cells. Cyan arrows: undifferentiated blastomeres. Red arrowhead: temporary embryonic pharynx. Red arrows: primitive gut cells. (B) S3 embryo stained with *EF1a-like-1* riboprobes (red) and sytox green nuclear counterstain (green). Cyan arrowheads: primitive ectoderm cells. Cyan arrows: undifferentiated blastomeres. Yellow arrowhead: temporary embryonic pharynx. (C) Confocal Z-slice of an ovary from a sexually mature *Smed* hermaphrodite stained with *piwi-1* riboprobes (green) and DAPI (blue). Yellow arrows: oocytes. (D) Dispersed cleavage. S2 embryo stained with *piwi-1* riboprobes (red, blastomeres) and antibodies raised against the mitotic epitope H3S10p (green). Nuclei stained with DAPI (blue). Yellow arrow: dividing blastomere. (E) *piwi-1* is expressed in undifferentiated blastomeres of S3 embryos. S3 embryo costained with riboprobes complementary to *piwi-1* (red) and *EF1a-like-1* (green). 100% *piwi-1+* blastomeres coexpressed *EF1a-like-1*. n = 159 cells scored, n = 5 S3 embryos. Cyan arrows: undifferentiated blastomeres. Yellow arrowhead: temporary embryonic pharynx. Red arrows: fiduciary beads used for SPIM reconstruction. (F) *piwi-1+* cells are located in the embryonic wall. Paraffin-embedded cross-section of a S3 sphere stained with *piwi-1* riboprobes (blue) and eosin (pink). Cyan arrows: *piwi-1+* cells. GC: yolk-filled gut cavity. Inset: magnified view of a *piwi-1+* cell. Scale: 25 μm. (G) Left: Average RPKM per embryo for *piwi-1* (S2–S8). Right: WISH developmental time course with *piwi-1* riboprobes (blue) (S2–S8). O, oral hemisphere; V, ventral. (A–G) Scale: 100 μm. Left: Observed distribution of *piwi-1+* cells in S3–S4 embryos (blue bars) relative to the oral-aboral axis (0–3.14 radians). Maximum likelihood analysis best described distribution by the function ((1-exp(-θ/θ'))*sin(θ), blue line). The optimal calculated θ' was 0.45 ± 0.045 radians, based on simulations with comparably sized data sets, and was several orders of magnitude more likely to explain the observed distribution than the theoretical normal distribution, sin(θ), (θ' = 0), red line. S3: n = 32 embryos, n = 1,746 *piwi-1+* cells scored. S4: n = 8 embryos, 2,665 *piwi-1+* cells scored. Right: observed *piwi-1+* cell distributions for individual S3 (top) and S4 (bottom) embryos. (C–G) *piwi-1+* cells are detected throughout embryogenesis. (H) *piwi-1+* cell positions are not stereotyped in S3–S4 embryos.

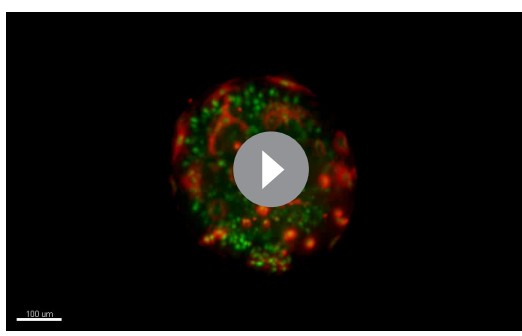

**Video 1.** S3 embryo architecture. SPIM reconstructed S3 embryo costained with *EF1a-like-1* (red) and sytox green nuclear counterstain. *EF1a-like-1* is a pan-embryonic cell marker that stains primitive ectoderm cells, the temporary embryonic pharynx and undifferentiated blastomeres in the embryonic wall. *EF1a-like-1* staining is absent from yolk cells in the embryonic wall and gut cavity.

*piwi-1* riboprobes and antibodies raised against the G2-M phase mitotic marker H3S10p showed that *piwi-1+* blastomeres divide asynchronously during S2 (*Figure 2D*).

As spheres form during S2, some blastomeres downregulate *piwi-1* expression and differentiate into temporary embryonic cell types. *piwi-1* expression was restricted to, and expressed throughout, the undifferentiated blastomere population in S3 embryos, as demonstrated by double fluorescent WISH with *EF1a-like-1* and *piwi-1* riboprobes (*Figure 2E*, *Video 2*). *piwi-1+* blastomeres were always located in the embryonic wall (*Figure 2F*). *piwi-1* expression was never detected in the primitive ectoderm, temporary embryonic pharynx or primitive gut (*Figure 2E, G*). During S3–S5, *piwi-1+* cell number clearly increased, effectively blanketing the sphere (*Figure 2G*). As definitive gut development proceeded during S6–S8, *piwi-1+* cells occupied the parenchyma between the developing gut branches (*Figure 2G*). Notably, *piwi-1+* cells were not detected in the definitive pharynx, and the compartment receded from the anterior margin as head structures developed (*Figure 2G*). By S8, the spatial distribution of *piwi-1+* cells was indistinguishable from that of the adult neoblast compartment (*Figure 2G*).

Analysis of *piwi-1+* cell positions in S3–S4 embryos revealed that undifferentiated blastomeres were spatially disordered, or 'anarchic,' as described for other freshwater flatworms (*Bardeen, 1902*; *Cardona et al., 2005*; *Hallez, 1887*; *Ijima, 1884*; *Le Moigne, 1963*; *Metschnikoff, 1883*; *Vara et al., 2008*). The observed distribution of *piwi-1+* cells in S3–S4 embryos with respect to the temporary embryonic pharynx (*Figure 2H*, blue line) was nearly identical to that of a random distribution (*Figure 2H*, red line). Fewer *piwi-1+* cells were located adjacent to the oral pole than was predicted for a theoretical normal distribution, perhaps due to spatial constraints imposed by the temporary embryonic pharynx and associated primitive gut cells (*Figure 2H*). We used a simple dampening term and maximum likelihood estimation to account for the deviation between the observed and theoretical normal distributions (*Figure 2H*). However, the distribution of *piwi-1+* cells in the embryonic wall was not stereotyped; it varied greatly across individuals (*Figure 2H*).

Strikingly, cell cycle activity was restricted to *piwi-1+* cells at all developmental stages assayed, and all *piwi-1+* cells in early embryos were cycling. Expression of the cell cycle regulators *PCNA* and *RRM2-2* closely mimicked that of *piwi-1* during embryogenesis, both with respect to the spatial distribution of positive cells and trends observed in the RNA-Seq data (*Figures 2G* and *3A–B*). Double fluorescent WISH on S3–S5 embryos revealed that *PCNA* and *RRM2-2* were expressed exclusively in *piwi-1+* blastomeres, all of which were cycling (*Figure 3C–D*, *Videos 3–4*). S3–S5 embryos costained with *piwi-1* and H3S10p antibodies confirmed that mitotic activity was restricted to

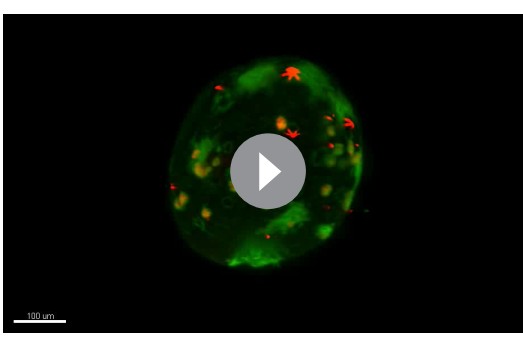

**Video 2.** *piwi-1* is expressed in all undifferentiated blastomeres of S3 embryos. SPIM reconstructed S3 embryo costained with *piwi-1* (red) and *EF1a-like-1* (green). *piwi-1* is expressed in all undifferentiated blastomeres in the embryonic wall (*piwi-1+, EF1a-like-1 +* cells). *piwi-1* is not expressed in differentiated tissues marked by *EF1a-like-1* alone, including the primitive ectoderm and temporary embryonic pharynx (green). Several fluorescent beads used for three-dimensional reconstruction are visible (red).

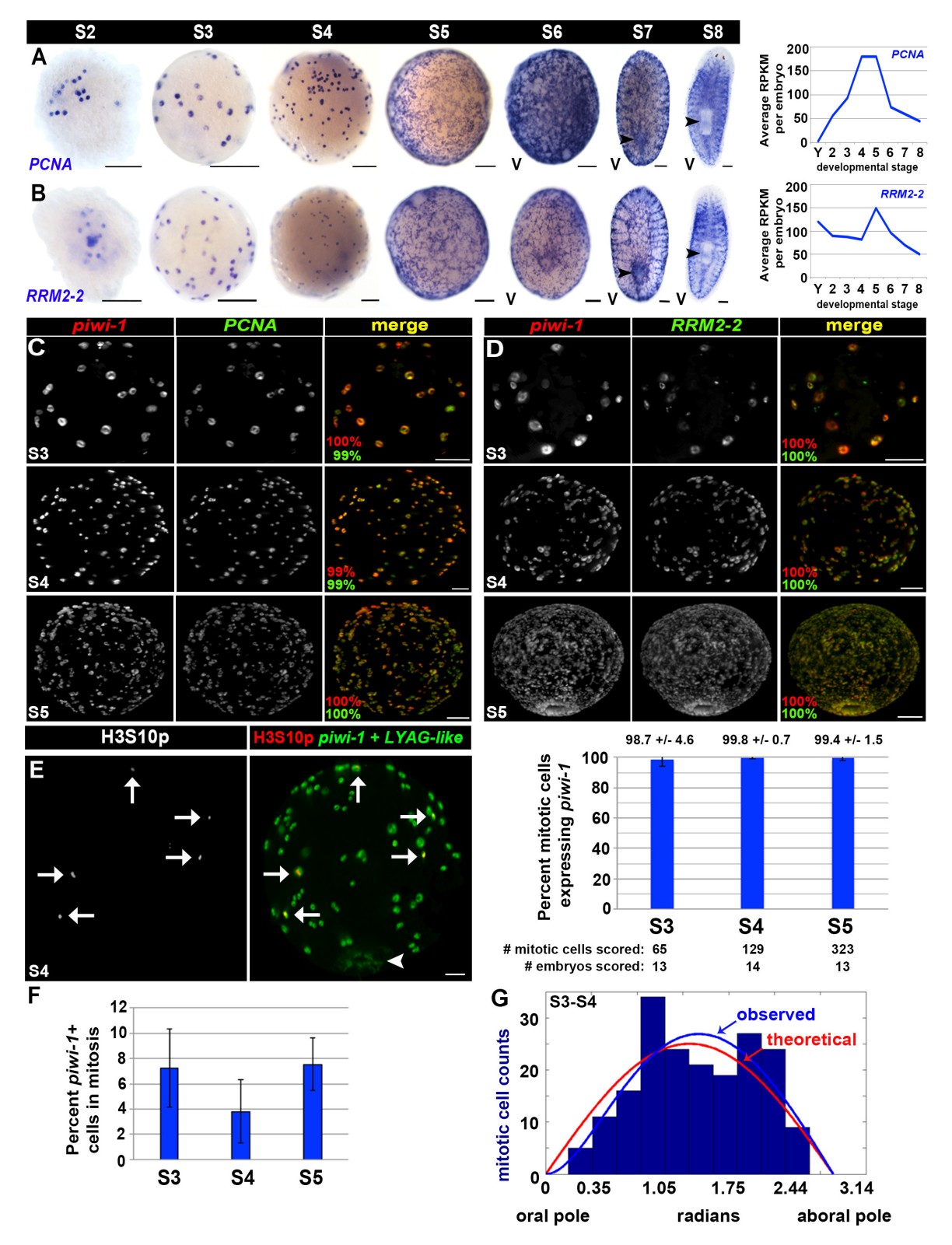

**Figure 3.** Cell cycle activity is restricted to the *piwi-1+* compartment. (A–B) Left: Colorimetric WISH depicting expression of *PCNA* (A) or *RRM2-2* (B) during stages S2–S8. Right: Average RPKM values per embryo for *PCNA* (A) or *RRM2-2* (B) in Y (yolk) and S2–S8. V, ventral. Scale: 100 μm. (C–D) S3 (top), S4 (middle) and S5 (bottom) embryos costained with *piwi-1* (red) and *PCNA* (green [C]) or *RRM2-2* (green [D]) riboprobes. The percentage of *piwi-1+* cells coexpressing the indicated cell cycle marker (red) and the percentage of *PCNA+* or *RRM2-2+* cells coexpressing *piwi-1* (green) appear in the

*Figure 3 continued on next page*

*Figure 3 continued*

lower left corner of merged images. Scale bars: 100 μm. (C) S3: n = 273 cells, n = 6 embryos. S4: n = 1,267 cells, n = 4 embryos. S5: n = 734 cells, n = 3 embryos. (D) S3: n = 130 cells, n = 4 embryos. S4: n = 1,295 cells, n = 5 embryos. S5: n = 350 cells, n = 3 embryos. (E) Mitotic activity is restricted to the *piwi-1+* cell compartment in S3–S5 embryos. Left: S4 embryo costained with *piwi-1* and the embryonic pharynx marker *LYAG-like* (both in green) and antibodies against the mitotic epitope H3S10p (red). White arrows: dividing blastomeres. White arrowhead: temporary embryonic pharynx. Scale bar: 100 μm. Right: Bar graph depicting the percentage of mitotic cells scored that expressed *piwi-1* in S3–S5 embryos. (F) The mitotic index for the *piwi-1+* cell compartment did not vary significantly during S3–S5. Average percentage of *piwi-1+* cells in mitosis during S3–S5. Error bars represent the standard deviation of the mean. Observed distribution of mitotic (*piwi-1+*, H3S10p+) cells in S3-–S4 embryos (blue bars) along the oral-aboral axis (0–3.14 radians). Using the function derived with maximum likelihood estimation for the *piwi-1+* cell distribution, $(1-exp(-\theta/\theta'))*sin(\theta)$ (blue line), and simulations using equivalent sample sizes, the optimal $\theta'$ was calculated to be $0.58 \pm 0.33$, and was 50-fold more likely to explain the observed trend than a simple normal distribution, $sin(\theta)$, where $\theta'=0$ (red line). S3: n = 82 mitotic cells, n = 18 embryos. S4: n = 110 mitotic cells, n = 8 embryos. (G) Mitotic cell positions are not stereotyped in early embryos.

*piwi-1+* blastomeres (*Figure 3E*, *Video 5*). Consistent with observations made in S2 embryos (*Figure 2D*), cell divisions were asynchronous during S3–S5. The mitotic index for S3–S5 *piwi-1+* blastomeres was stable, with no statistically significant difference in the calculated division rate (*Figure 3F*). Analysis of mitotic (*piwi-1+*, H3S10p+) cell positions along the oral-aboral axis in S3–S4 embryos did not reveal regional biases in mitotic activity across samples (*Figure 3G*).

Several striking parallels may be drawn regarding cell cycle behavior of the *piwi-1+* population during embryogenesis and adulthood. First, cell cycle activity is largely restricted to this compartment in both contexts, with exception of the male and female germline in sexually mature adults (*Baguñà, 1976*; *Newmark and Sánchez Alvarado, 2000*; *Reddien et al., 2005*; *Wagner et al., 2011*; *Wang et al., 2007*). Second, cycling cells and mitotic figures do not display obvious positional biases within the parenchyma in either S3–S5 embryos or adults during homeostasis (*Newmark and Sánchez Alvarado, 2000*). Little substantiating evidence exists in support of a quiescent *piwi-1+* cell population during embryogenesis or adulthood (*Newmark and Sánchez Alvarado, 2000*), and differences in cell cycle length were not observed between neoblast subclasses in adult asexual animals (*van Wolfswinkel et al., 2014*). Finally, the staggering net increase in *piwi-1+* blastomeres during S3–S5 suggests a capacity for self-renewal, a property possessed by neoblasts.

## Many adult asexual stem cell genes are expressed throughout embryonic development

Numerous neoblast-enriched transcripts, identified through whole asexual animal irradiation studies and cell sorting (*Eisenhoffer et al., 2008*; *Labbé et al., 2012*; *Rossi et al., 2007*; *Solana et al., 2012*; *Wagner et al., 2012*; *Wurtzel et al., 2015*), have been vetted for co-expression with *piwi-1+* and ascribed function(s) in neoblast proliferation, maintenance or cell fate commitment. Some neoblast-enriched transcripts, including those encoding nuage components and cell cycle regulators, are expressed in all neoblasts, whereas others are predominantly expressed in subpopulation(s) of cells that may be primed to adopt differentiated fates (*van Wolfswinkel et al., 2014*). To address similarities in the gene expression profiles of the *piwi-1+* population during embryogenesis and adulthood, the expression trends of 242 adult asexual neoblast-enriched transcripts were examined using the molecular staging series data. Neoblast-enriched transcript membership was determined by sequences downregulated in whole animals following lethal irradiation across three independent experiments (*Duncan et al., 2015*; *Wagner et al., 2012*) (Chen and Sánchez Alvarado, personal

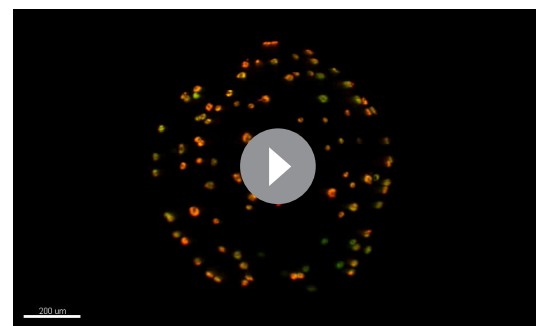

**Video 3.** Cell cycle activity is restricted to *piwi-1* blastomeres, and all blastomeres are cycling. SPIM reconstructed S4 embryo costained with *piwi-1* (red) and *PCNA* (green). *PCNA* expression is restricted to *piwi-1+* blastomeres, and all *piwi-1+* cells co-express *PCNA*.

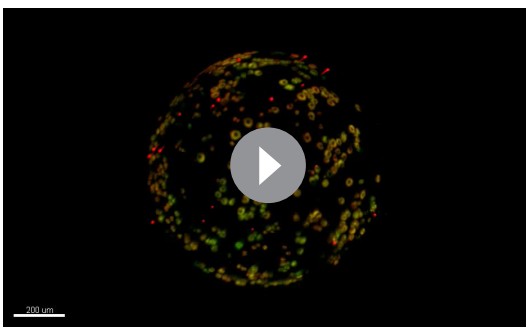

**Video 4.** Cell cycle activity is restricted to *piwi-1* blastomeres, and all blastomeres are cycling. SPIM reconstructed S4 embryo costained with *piwi-1* (red) and *RRM2-2* (green). *RRM2-2* expression is restricted to *piwi-1+* blastomeres, and all *piwi-1+* cells co-express *RRM2-2*. Several fluorescent beads used for three-dimensional reconstruction are visible (red).

**Video 5.** Mitotic activity is restricted to *piwi-1+* blastomeres, which cycle asynchronously. SPIM reconstructed S4 embryo costained with *piwi-1* and *LYAG-like* (both in green) and H3S10p antibodies (red). *LYAG-like* marks the temporary embryonic pharynx and is not expressed in *piwi-1+* blastomeres. Several examples of *piwi-1+*, H3S10p+ cells are evident.

communication). Strikingly, most adult asexual neoblast-enriched transcripts were expressed throughout embryogenesis (*Figure 4A*, *Figure 4—source data 1*). 74% (n = 180) of the neoblast-enriched transcripts had average RPKM values per embryo ≥1.0 in S2 embryos and 52% (n = 128) transcripts had five-fold or greater expression levels in S2 embryos versus yolk, raising the possibility that other adult stem cell genes were expressed in blastomeres. Consistent with this idea, 41% of the adult asexual neoblast-enriched transcripts (n = 99) were present in the molecular expression signature(s) for S2–S5 embryos (*Figure 1—source data 2–5*, *Figure 4—source data 1*). Expression of neoblast-enriched transcripts usually peaked during S4 or S5, prior to construction of the definitive organ systems, and diminished thereafter (*Figure 4A*, *Figure 4—source data 1*). The apparent decrease in expression after S5 was likely attributable to drastic changes in the complexity of the single embryo RNA samples during organogenesis, and was similarly observed for *piwi-1* and the cell cycle regulators *PCNA* and *RRM2-2* (*Figures 2G* and *3A–B*, *Figure 4—source data 1*).

Hierarchical clustering of the 242 adult asexual neoblast-enriched transcripts revealed correlated expression of genes associated with DNA replication (e.g., the replication licensing factors *MCM2* and *MCM5*), DNA repair (e.g., *fancd2-like*, *msh2* and *msh6*) and cell cycle progression (e.g., *cyclin D-like*, *cyclin-B1* and *cyclin-B2*) during embryogenesis, and the expression trends for these genes mimicked those of *PCNA* and *piwi-1* (*Figure 4A* [Cluster 1], *Figure 4—source data 1*). Notably, transcripts encoding the nuage-associated factors *piwi-2* and *piwi-3*, the RNA-binding protein *bruli-1*, and the transcription factors *SoxP-1* and *junL1-1*, all genes previously implicated in neoblast maintenance or function (*Guo et al., 2006*; *Palakodeti et al., 2008*; *Reddien et al., 2005*; *Wagner et al., 2012*), were also coregulated with *piwi-1* during embryogenesis (*Figure 4A* [Cluster 3], *Figure 4—source data 1*). Consistent with the expression trends detected by RNA-Seq and the previously reported expression pattern for *Schmidtea polychroa* (*Spol*) *tud-1* (*Solana et al., 2009*), *piwi-2*, *piwi-3*, *tud-1*, *bruli-1*, *Sox-P1* and *Sox-P2* were expressed in cells with similar morphology and distribution to the *piwi-1+* population during embryogenesis (*Figure 4B–E*, *Figure 4—figure supplement 1A–C*). Double fluorescent WISH revealed coincident expression of *piwi-1* and the adult stem cell markers *piwi-2*, *piwi-3*, *tud-1* and *bruli-1* in S3–S5 embryos (*Figure 4F–I*, *Figure 4—figure supplement 1D–F*, *Videos 6–9*). Y12 antibodies, which label chromatoid bodies in adult neoblasts (*Rouhana et al., 2012*), stained *piwi-1+* blastomeres during S4-–S5 (*Figure 4—figure supplement 1G*, *Video 10*). Taken together, these findings suggest that many adult asexual neoblast markers, including genes implicated in DNA replication and repair, cell cycle control, chromatin remodeling and/or modification, genome surveillance and pluripotency, are also likely to be expressed throughout the *piwi-1+* population during embryogenesis. Moreover, shared elements of the blastomere and neoblast expression signatures, including DNA replication and repair pathway, cell cycle, nuage

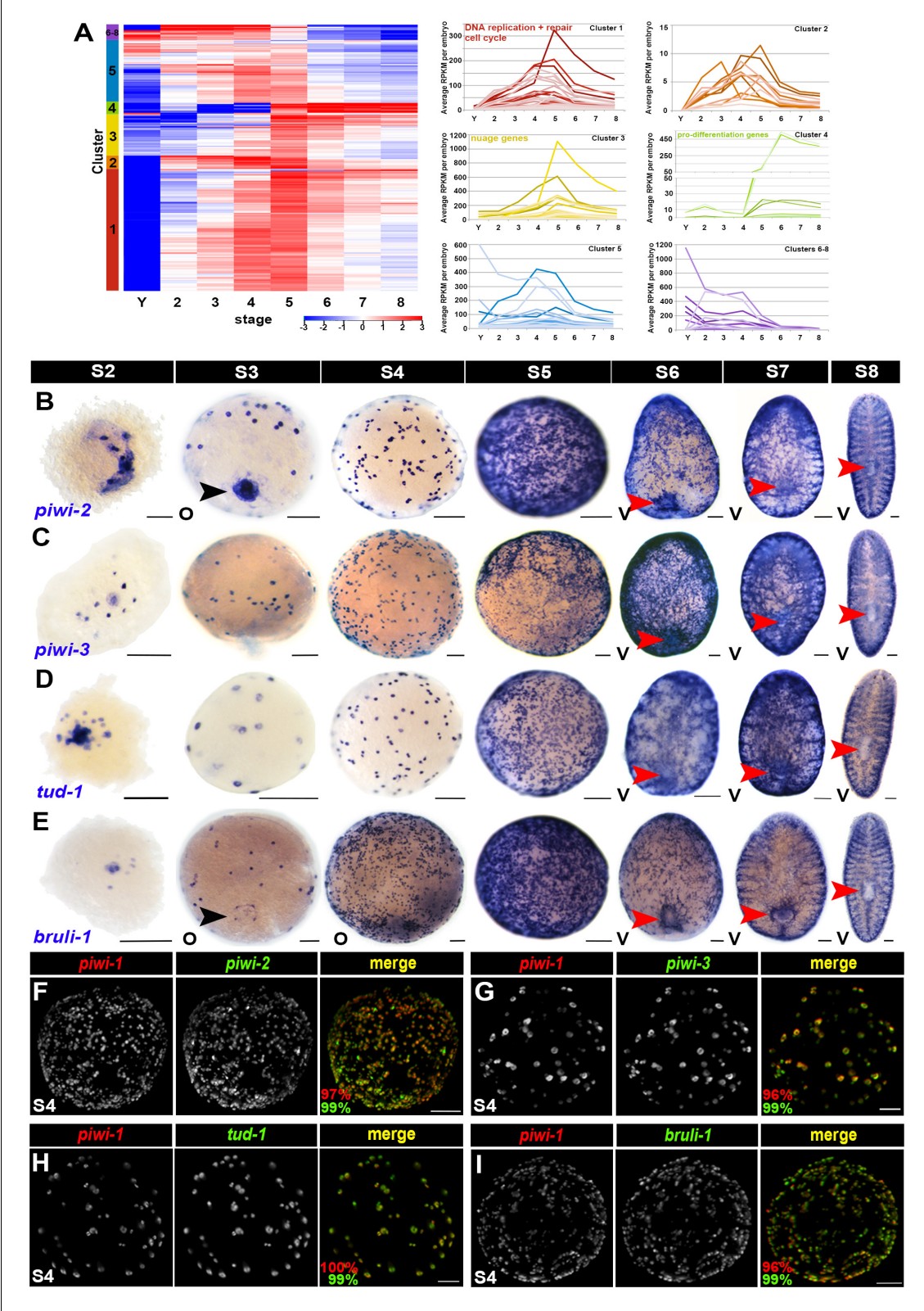

**Figure 4.** Many adult neoblast markers are similarly expressed throughout the *piwi-1+* compartment during embryogenesis. (**A**) Many transcripts with adult asexual neoblast-enriched expression are expressed throughout embryogenesis. Hierarchical clustering of 242 adult asexual neoblast-enriched transcripts during embryonic development using normalized mixed stage reference comparison data. Left: Heat map. Colored bars (left) denote clusters. Right: Normalized average RPKM values per embryo, plotted as a function of developmental time, for Clusters 1–8. Y, yolk; S2–S8, Stages 2–8.
*Figure 4 continued on next page*

*Figure 4 continued*

(B–E) Colorimetric WISH depicting expression of *piwi-2* (B), *piwi-3* (C), *tud-1* (D) and *bruli-1* (E) during embryogenesis (blue) (S2–S8). V, ventral. Black arrowheads: temporary embryonic pharynx. Red arrowheads: definitive pharynx. Scale bars: 100 μm. (F–I) Many markers of the adult asexual neoblast compartment are also expressed in *piwi-1+* blastomeres. Fluorescent WISH on S4 embryos with riboprobes against *piwi-1* (red) and *piwi-2* (F), *piwi-3* (G), *tud-1* (H) or *bruli-1* (I) (green). Percentage of *piwi-1+* cells coexpressing the indicated marker (red) and the percentage of the indicated adult asexual neoblast marker coexpressing *piwi-1* (green) appears in the lower left corner of merged images. Scale bars: 100 μm. (F) n = 435 cells, n = 9 S3–S4 embryos. (G) n = 535 cells, n = 5 S3–S4 embryos. (H) n = 1,867 cells, n = 8 S3–S5 embryos. (I) n = 1,353 cells, n = 3 S4 embryos.

The following source data and figure supplement are available for figure 4:

**Source data 1.** Hierarchical clustering analysis for 242 adult asexual neoblast-enriched transcripts.
**Figure supplement 1.** Adult asexual neoblast-enriched markers are coexpressed in *piwi-1+* cells during embryogenesis.

and RNA processing genes, are prominent features of an evolutionarily conserved gene expression signature for metazoan primordial stem cells (*Alié et al., 2015*).

## Early-embryo-enriched transcripts expressed throughout *piwi-1+* blastomeres are downregulated as organogenesis begins

Hierarchical clustering of S2-–S4-enriched transcripts using scaled RPKM values identified 1,048 sequences in Clusters 5, 6 and 8 that were downregulated by S5 and remained lowly expressed through S8; these sequences are referred to as early-embryo-enriched (EEE) transcripts (*Figure 5A*, *Figure 5—source data 1*). EEE transcripts were likely expressed in blastomeres and/or temporary embryonic tissues, as 98% of the sequences had average expression values at least five-fold greater in S2 embryos than in Y (*Figure 5—source data 1*). Most EEE transcripts were expressed at low levels in intact adults regardless of biotype: average RPKM values less than 1.0 were recorded for 65% and 59% of the EEE transcripts in C4 or SX, respectively (*Figure 5—source data 1*).

EEE transcript expression trends during embryonic development and adulthood were validated using the Nanostring nCounter platform (*Geiss et al., 2008*). Total RNA replicates from single S2–S8 embryos, Y, C4 and SX adults were queried for expression of 108 EEE transcripts using a custom probe set (Materials and methods, *Figure 5—source data 2*). Curiously, S2 replicates had the lowest summed read counts across the experiment (*Figure 5—figure supplement 1A*), and expression of *piwi-1*, the cell cycle marker *H2B*, and EEE transcripts was not detected during S2 (*Figure 5 – Figure 1—figure supplement 1E–F*, *Figure 5—source data 2*), perhaps due to lack of template amplification in the nCounter assays. Apart from the S2 samples, expression trends for *piwi-1*, *H2B*, the pro-differentiation factor *prog-1* and more than 90% of the EEE transcripts mirrored those observed in the RNA-Seq time course (*Figure 5—figure supplement 1E–F*). Most EEE transcripts queried on the nCounter platform showed peak expression during S3 and/or S4, downregulation during S5, and low to undetectable expression in late stage embryos, C4 or SX adults (*Figure 5—figure supplement 1F*). Hierarchical clustering using normalized Nanostring count data identified three cohorts of EEE transcripts that hadcomposition and decay kinetics similar to those observed in the RNA-Seq time course; 65% of the EEE transcripts assayed co-clustered in both the RNA-Seqand nCounter analyses (*Figure 5—figure supplement 1F*).

To determine which cell types express EEE transcripts, and to examine spatiotemporal

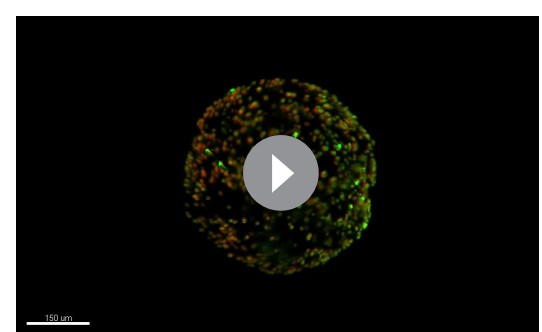

**Video 6.** *piwi-1+* blastomeres co-express the adult asexual neoblast-enriched gene *piwi-2*. SPIM reconstructed S4 embryo costained with *piwi-1* (red) and *piwi-2* (green). *piwi-1+* blastomeres co-express the nuage factor *piwi-2*, and virtually all *piwi-2+* cells co-express *piwi-1*. Several fluorescent beads used for three-dimensional reconstruction are visible (green).

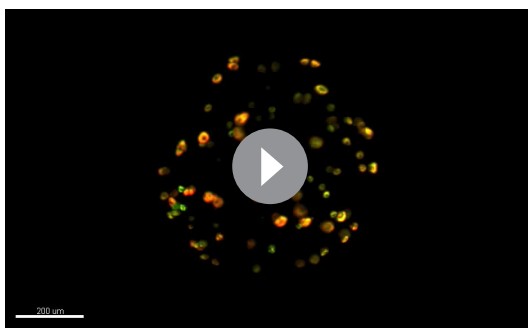

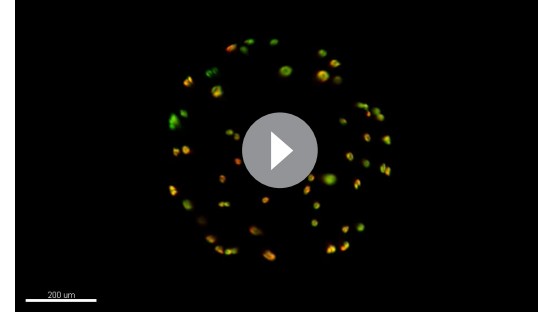

**Video 7.** *piwi-1+* blastomeres co-express the adult asexual neoblast-enriched gene *piwi-3*. SPIM reconstructed S4 embryo costained with *piwi-1* (red) and *piwi-3* (green). *piwi-1+* blastomeres co-express the nuage factor *piwi-3*, and virtually all *piwi-3+* cells co-express *piwi-1*.

**Video 8.** *piwi-1+* blastomeres co-express the adult asexual neoblast-enriched gene *tud-1*. SPIM reconstructed S4 embryo costained with *piwi-1* (red) and *tud-1* (green). *piwi-1+* blastomeres co-express the nuage factor *tud-1*, and virtually all *tud-1+* cells co-express *piwi-1*.

changes in EEE transcript expression, colorimetric WISH was performed on S2–S8 embryos and intact C4 adults (*Figure 5—source data 3*). Some EEE transcripts were expressed exclusively in differentiated temporary embryonic tissues. *VAL-like*, *MPEG1-like-1*, *MPEG1-like-2* and *netrin-like* were solely expressed in the temporary embryonic pharynx until S6, whereas *gelsolin-like* and *4XLIM-like* were expressed in both the primitive ectoderm and temporary embryonic pharynx during S3–S4 (*Figure 5—source data 3*, *Figure 1—figure supplements 12C–G* and *11D–E*). Expression of these EEE transcripts is likely under zygotic control, occurring during or after the cell fate decisions to downregulate *piwi-1*, exit the cell cycle and differentiate.

Most EEE transcripts queried by WISH (n = 15, 75% assayed) were expressed in both undifferentiated blastomeres and temporary embryonic tissue(s) during S3–S4 (*Figure 5B–E*, *Figure 5—source data 3*). Some of these transcripts may be maternally deposited, albeit we cannot ascertain the relative contribution(s) of maternal and zygotic expression from our RNA-Seq data. Expression of these EEE transcripts diminished greatly by S5, with EEE transcripts that had been expressed at moderate or low levels becoming undetectable; specific expression of robustly expressed transcripts sometimes persisted until S6 (*Figure 5B–E*, *Figure 5—source data 3*). Consistent with the RNA-Seq and Nanostring nCounter results, EEE transcript expression was not detected by colorimetric WISH in S7 or S8 embryos or in C4 adults (*Figure 5B–E*, *Figure 5—source data 3*). Fluorescent double WISH performed with riboprobes complementary to *piwi-1* and the EEE transcripts *tct-like*, *BTF3-like*, *DDX5-like* and *eIF4a-like* revealed coincident expression throughout the S4 blastomere compartment (*Figure 5F–I*). Intriguingly, EEE transcript expression often decayed quicker in differentiated cells than in undifferentiated blastomeres, raising the possibility that regulation of EEE transcription and/or transcript stability may vary by cell type. Robust expression of many EEE transcripts was detected in blastomeres during S3–S4, EEE transcript expression in temporary embryonic tissues was present during S3 and drastically diminished by S4 (*Figure 5B–E*, *Figure 5—source data 3*).

EEE transcripts expressed throughout the undifferentiated *piwi-1+* blastomere population

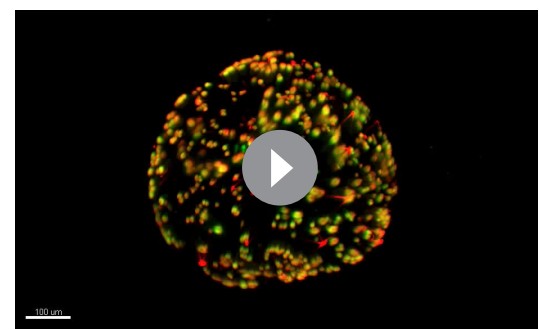

**Video 9.** *piwi-1+* blastomeres co-express the adult asexual neoblast-enriched gene *bruli-1*. SPIM reconstructed S4 embryo costained with *piwi-1* (red) and *bruli-1* (green). *piwi-1+* blastomeres co-express the stem cell maintenance gene *bruli-1*, and virtually all *bruli-1+* cells co-express *piwi-1*. Several fluorescent beads used for three-dimensional reconstruction are visible (red).

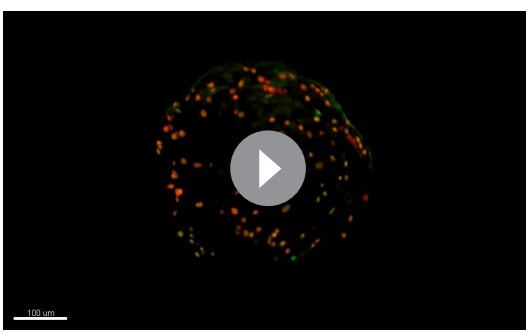

**Video 10.** *piwi-1+* blastomeres possess chromatoid bodies. SPIM reconstructed S4 embryo costained with *piwi-1* (red) and Y12 antibodies (green). Y12 antibody staining was restricted to, and present throughout, the *piwi-1+* blastomere population.

in S3–S4 embryos are downregulated as definitive organogenesis begins during S5. These transcripts likely represent a key temporal shift in the expression profile of *piwi-1+* cells during embryogenesis. Moreover, EEE transcript expression provides a molecular metric to distinguish *piwi-1+* blastomeres from adult neoblasts.

## Adult lineage progenitors arise within *piwi-1+* blastomeres as organogenesis begins

Many developmental regulators implicated in lineage commitment and differentiation were expressed at low levels in S2–S4 embryos, and were upregulated dramatically as definitive organogenesis began during S5. Key regulators of cell fate specification for many tissues, including the epidermis (*p53* and zfp-1) (*Pearson and Sánchez Alvarado, 2010*; *van Wolfswinkel et al., 2014*; *Wagner et al., 2012*), nervous system (*coe, sim, pax3/7 like, lhx1/5–1* and *pitx*) (*Cowles et al., 2013*; *Currie and Pearson, 2013*; *März et al., 2013*; *Scimone et al., 2014*), excretory system (*pou2/3, six1/2–2, eya, sal1* and *osr*) (*Scimone et al., 2011*), photoreceptor neurons (*eya, six-1/2, otxA* and *soxB*), pigment cup cells (*eya, six-1/2, sp6-9* and *dlx*) (*Lapan and Reddien, 2011, 2012*) and primordial germ cells (*nos*) (*Wang et al., 2007*) were among the S5- and/or S6-enriched transcripts (*Figure 1—source data 5–6*). Additional validated or putative drivers of cell fate determination in muscle (*myoD*) (*Cowles et al., 2013*; *Scimone et al., 2014*), the gastrovascular system (*prox-1* and *foxA1*) (*Adler et al., 2014*; *Scimone et al., 2014*; *van Wolfswinkel et al., 2014*), nervous system (*lhx2/9, six3-1, nkx6-like, otxB-like, otxA, pax6a* and *pax6b*) (*Pineda et al., 2002*; *Scimone et al., 2014*) and eyes (*ovo*) (*Lapan and Reddien, 2012*) showed statistically significant upregulation during S5 and/or S6, albeit with adjusted p-values and fold-changes above the stringent thresholds set for inclusion in the S5–S6-enriched transcript lists. GO analysis on S5-enriched transcripts showed statistically significant enrichment of terms associated with patterning and cell fate specification, transcriptional regulation, and development of organ systems including the epidermis, central and peripheral nervous system, muscle, digestive and excretory systems (*Figure 1—source data 5*). Taken together, these observations suggest that formation of many definitive organ systems begins during S5, a supposition bolstered by WISH developmental time course data depicting expression patterns for numerous progenitor and cell type-specific markers during embryogenesis (*Figure 1—source data 9*, *Figure 1—figure supplements 13–19*).

Adult asexual knockdown phenotypes for many developmental TFs upregulated during S5 and S6 suggest that these genes are required for lineage specification, tissue maintenance, and production of new tissue during regeneration, with correspondence between affected tissues and site(s) of expression. Heterogeneous expression of TFs in neoblasts and post-mitotic progenitors informed the hypothesis that the neoblast population contains pluripotent stem cells as well as cycling, lineage-primed progenitors (*Reddien, 2013*). Single cell sequencing (SCS) studies suggest that the zeta (ζ) and gamma (γ) neoblast subclasses are epidermal and gut progenitors, respectively (*van Wolfswinkel et al., 2014*; *Wurtzel et al., 2015*), while the sigma (σ) neoblast subclass likely contains both pluripotent stem cells and progenitors for other lineages, including neural subtypes, protonephridia and primordial germ cells (*van Wolfswinkel et al., 2014*). In practice, coexpression of pan-neoblast markers (e.g., *piwi-1*) and developmental TFs is used for neoblast subclass identification.

*Smed* embryos are wholly reliant on cycling *piwi-1+* cells for creation of new tissues, and heterogeneous expression of key developmental TFs within *piwi-1+* blastomeres is predicted to generate the diverse array of progenitors required for organogenesis. While only a small fraction of the *piwi-1 +* compartment is predicted to be double positive for any given lineage marker, the entire population of lineage-positive cells is predicted to be positive for *piwi-1* at its inception. As development

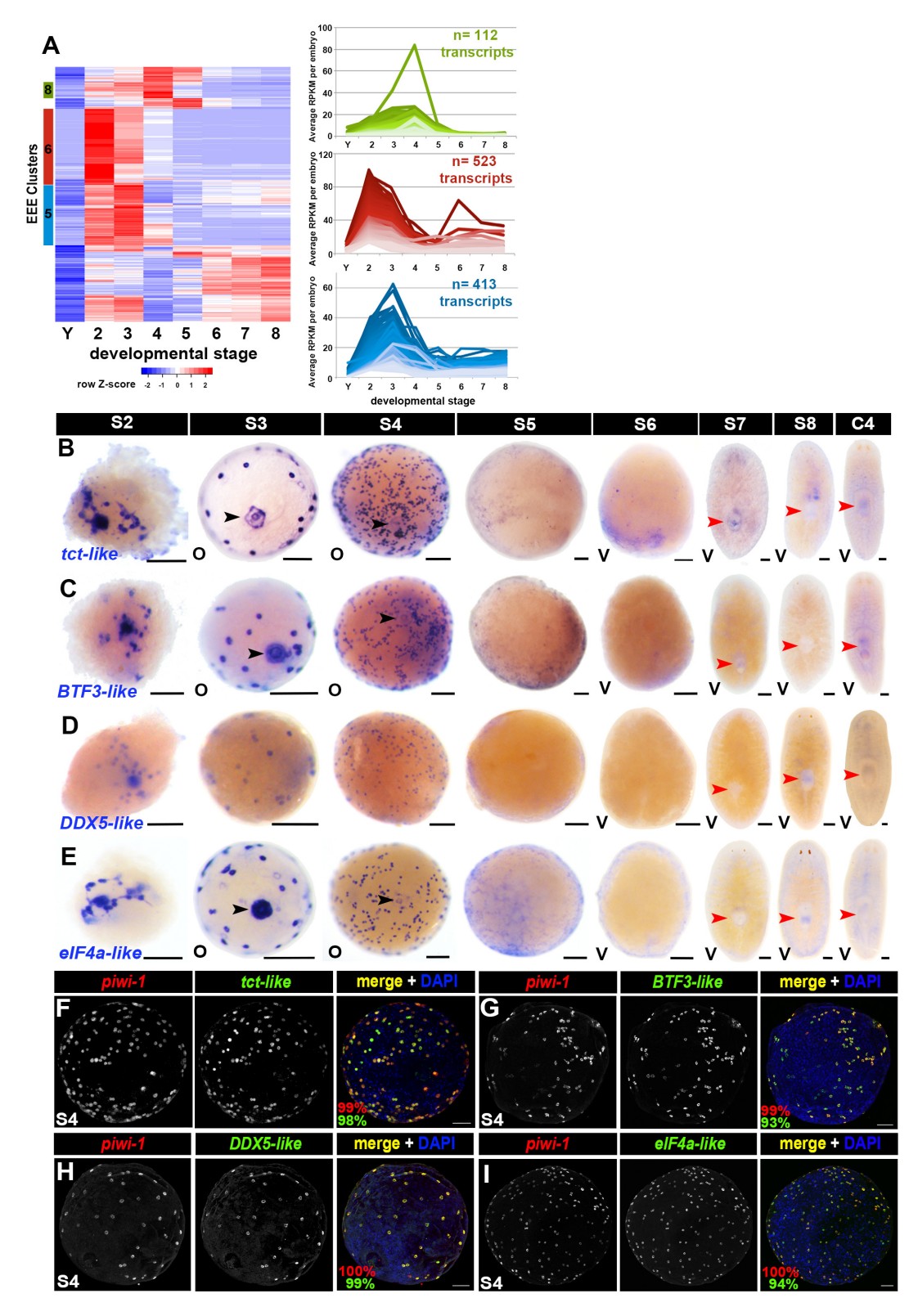

**Figure 5.** Early-embryo-enriched transcripts are downregulated as organogenesis begins. (**A**) Hierarchical clustering of S2–S4-enriched transcripts (n = 1,756) using scaled RPKM data. Left: Heat map. Y, yolk. Colored bars (left) denote Clusters 5, 6 and 8 containing early-embryo-enriched (EEE) transcripts. Cluster f5 sequences (blue, n = 413) were expressed at roughly equivalent levels during S2 and S3, with 66% (n = 275) transcripts showing five-fold or greater declines in average expression values between S3 and S5. Cluster 6 sequences (red, n = 523) exhibited maximal expression during

*Figure 5 continued on next page*

*Figure 5 continued*

S2, and average expression levels declined more than five-fold between S2 and S4 for 81% (n = 426) of these transcripts. Cluster 8 sequences (green, n = 112) showed peak expression during S4, with 52% (n = 60) of the transcripts showing five-fold or greater declines in average expression values by S5. Right: Normalized expression trends for EEE transcripts in Clusters 5 (blue), 6 (red) and 8 (green) plotted as a function of developmental time. Median 50% of transcripts based on expression maxima are plotted. (B–E) Colorimetric WISH depicting expression of the EEE transcripts *tct-like* (B), *BTF3-like* (C), *DDX5-like* (D) and *eIF4a-like* (E) (blue) in S2–S8 embryos and C4 asexual adults. Black arrowheads: temporary embryonic pharynx. Red arrowheads: definitive pharynx. O, oral; V, ventral. Scale bars: 100 µm. (F–I) EEE transcripts were expressed throughout the *piwi-1+* compartment in S3–S4 embryos. Fluorescent double WISH with riboprobes against *piwi-1* (red) and the EEE transcripts *tct-like* (F), *BTF3-like* (G), *DDX5-like* (H) and *eIF4a-like* (I) (green) in S4 embryos. Percentage *piwi-1+* cells coexpressing the indicated EEE marker (red) and percentage EEE+ cells coexpressing *piwi-1* (green) appear in the lower left corner of merged images. (F) n = 895 *piwi-1+* cells, n = 905 *tct-like+* cells, n = 7 S3–S4 embryos. (G) n = 692 *piwi-1+* cells, n = 728 *BTF3+* cells, n = 6 S3–S4 embryos. (H) n = 676 *piwi-1+* cells, n = 681 *DDX5-like+* cells, n = 5 S3–S4 embryos. (I) n = 312 *piwi1+* cells, n = 332 *eIF4a+* cells, n = 4 S3–S4 embryos.

The following source data and figure supplement are available for figure 5:

**Source data 1.** Hierarchical clustering of S2–S4-enriched transcripts across embryogenesis.
**Source data 2.** Validation of transcript expression trends using the Nanostring nCounter platform.
**Source data 3.** EEE transcript expression patterns detected by colorimetric WISH.
**Figure supplement 1.** Validation of early-embryo-enriched transcript expression trends using the Nanostring nCounter platform.

proceeds, the fraction of lineage-positive cells coexpressing *piwi-1* will decrease as cells downregulate expression of *piwi-1* and differentiate further.

If parallels with neoblasts hold true, then *piwi-1+* blastomeres should self-renew and give rise to differentiating progeny during S5. To assay for cells exiting the *piwi-1+* compartment during S5, embryos were costained with *piwi-1* riboprobes and PIWI-1 antibodies. In adults, *piwi-1* mRNA is restricted to the neoblast population, whereas PIWI-1 protein perdures in early post-mitotic progeny (*Guo et al., 2006*; *Scimone et al., 2011*; *Wagner et al., 2011*). Indeed, recent work suggests that mechanisms exist to sequester *piwi-1* mRNA and chromatoid bodies within one daughter cell during neoblast division, producing one cell that maintains neoblast identity and one cell that differentiates (*Lei et al., 2016*). Virtually all S5 *piwi-1+* blastomeres also contained PIWI-1 protein (*Figure 6—figure supplement 1*). Rare cells positive for PIWI-1 protein but in which *piwi-1* mRNA was undetectable were observed in S5 embryos, suggesting that some of the division progeny exited the *piwi-1+* blastomere population (*Figure 6—figure supplement 1*).

To address whether lineages required for organogenesis arise within *piwi-1+* blastomeres during S5, expression of four evolutionarily conserved TFs implicated in tissue differentiation across three germ layers was examined singly and in combination with *piwi-1*. *p53* and *pax6a*, regulators of epidermal and neural fates, respectively, were proxies for ectodermal derivatives. Populations expressing *myoD*, a master regulator of muscle fate, were considered mesodermal derivatives, and populations expressing *gata456a*, a regulator of gut development, represented endodermal derivatives. While expression of these TFs was occasionally detected in a small number of cells during S4, robust expression of *p53* (*Figure 1—figure supplement 15B*), *gata456a* (*Figure 1—figure supplement 13F*), *myoD* (*Figure 1—figure supplement 17A*) and *pax6a* (*Figure 1—figure supplement 16B*) manifest in scattered parenchymal cells during S5. Indeed, fluorescent double WISH with *piwi-1* and *p53*, *gata456a*, *myoD* or *pax6a* identified prospective epidermal, gut, muscle and neural progenitor populations that coexpressed *piwi-1* and the developmental TFs in S5 embryos, as well as single positive populations for all of the markers assayed (*Figure 6A–D*, *Videos 11–14*).

The advent of developmental TF expression within *piwi-1+* blastomeres during S5, coupled with downregulation of EEE transcripts, may signal the emergence of molecularly distinct subpopulations akin to the neoblast subclasses. Whole embryo expression trends for EEE, σ, γ and ζ class transcripts suggest that large scale, cell-intrinsic shifts in gene expression occur within blastomeres between S4 and S5. Moreover, the developmental output of *piwi-1+* blastomeres, as described by molecular fate mapping, diversifies greatly during S5 and S6.

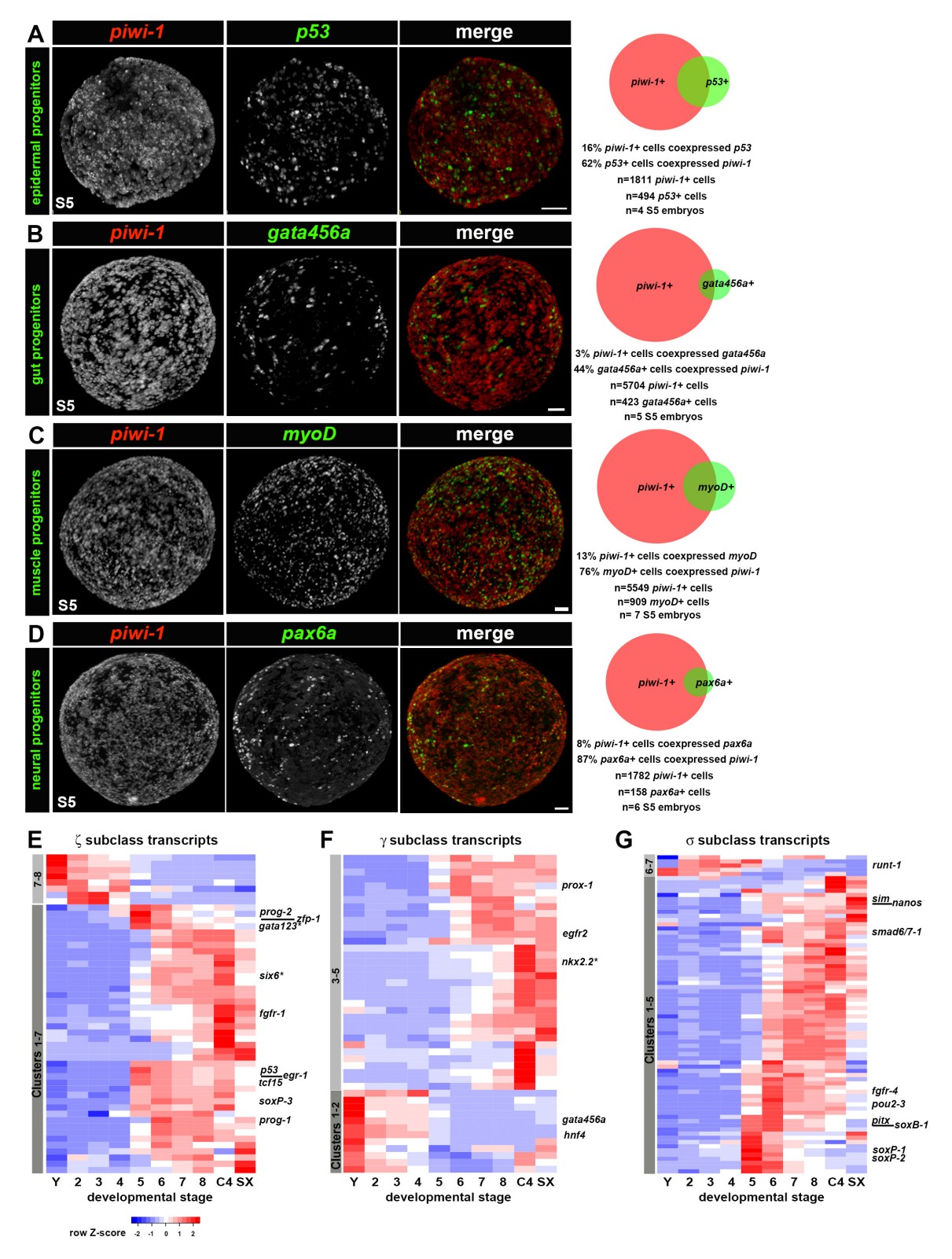

**Figure 6.** Adult lineages arise within the *piwi-1+* blastomere population as organogenesis begins. (A–D) Developmental transcription factors implicated in tissue specific differentiation programs are expressed in subpopulations of *piwi-1+* cells during S5. Fluorescent WISH with *piwi-1* (red) and *p53* (A), *gata456a* (B), *myoD* (C) and *pax6a* (D) (green) riboprobes on S5 embryos. Embryos in (B-D) were costained with *VAL*-like, a temporary embryonic pharynx specific marker (also in red). Right: Venn diagrams depict percentages of cells that were single or double positive for *piwi-1* and the indicated
*Figure 6 continued on next page*

*Figure 6 continued*

TFs. Scale bars: 100 µm. (**E–G**) Hierarchical clustering of zeta (ζ, **E**), gamma (γ, **F**) and sigma (σ, **G**) neoblast subclass-enriched transcripts during embryogenesis (Y and S2–S8), and in asexual (C4) and virgin sexual (SX) adults.
The following source data and figure supplement are available for figure 6:

**Source data 1.** Behavior of ζ, γ and σ adult asexual neoblast subclass-enriched transcripts during embryogenesis.
**Figure supplement 1.** PIWI-1 protein may perdure in cells committed to differentiation.

The expression signatures of the ζ and γ neoblast subclasses emerge during S5. ζ neoblasts require *p53* and *zfp-1* activities for production of post-mitotic mesenchymal progenitors, which simultaneously differentiate, migrate and ultimately integrate into the epidermis during homeostasis (*van Wolfswinkel et al., 2014*). *p53*, *zfp-1*, *tcf15*, *sox-P3*, *fgfr1*, *egr-1*, *six6\**, *gata123\** (*van Wolfswinkel et al., 2014*) and 15 additional transcripts used as classifiers of ζ subgroup identity in SCS experiments (*Wurtzel et al., 2015*) displayed statistically significant upregulation in whole embryos during S5 or later developmental stages (*Figure 6E*, *Figure 6—source data 1*, *Figure 1—figure supplement 15A*, *Figure 1—source data 5–8*). Furthermore, the ζ neoblast transcripts *soxP3, egr-1, fgfr1* and *prog-1* clustered together in the analysis of adult asexual neoblast-enriched transcripts, indicating that they displayed similar expression profiles during embryogenesis (*Figure 4A*, *Figure 4—source data 1*). Transcripts specifically expressed in post-mitotic epidermal progenitors downstream of ζ neoblasts, including *prog-1*, *AGAT-1* and *zpuf-6*, were enriched and first detected by WISH during S5 (*Figure 1—figure supplement 15C,E–G*, *Figure 1—source data 5*). γ neoblasts are identified by enriched expression of *gata456a*, *hnf4*, *prox-1* and *nkx2.2\** (*van Wolfswinkel et al., 2014*; *Wagner et al., 2011*). *prox-1* and *nkx2.2* were expressed at low levels in S2–S4 embryos and showed statistically significant upregulation during S5, and 15 additional γ neoblast-enriched transcripts identified in SCS experiments were enriched during S5 or later in development (*Figure 6F*, *Figure 6—source data 1*, *Figure 1—figure supplement 13E*, *Figure 1—source data 5–8*). The molecular staging series detected expression of *gata456a* and *hnf4* in yolk and early embryos (S2–S4) (*Figure 6F*). However, *gata456a* and *hnf4* expression were solely detected in the embryo proper by WISH, first in the developing temporary embryonic pharynx during S2 and later in parenchymal cells during S5 (*Figure 1—figure supplement 13F–G*).

Charting the emergence of the σ neoblast subclass is hampered by limitations of the subclass designation: σ neoblasts are presumed to be an amalgamation of several progenitor populations and pluripotent stem cells. Several σ -class genes, notably *SoxP-1* and *SoxP-2*, were expressed in parenchymal cells throughout embryogenesis, similar to *piwi-1* (*Figure 4A*, *Figure 4—source data 1*, *Figure 4—figure supplement 1A–C*, *Figure 6—source data 1*). *SoxP-1* is required for neoblast maintenance (*Wagner et al., 2012*), suggesting that σ-class transcripts with expression profiles to *SoxP-1* and *SoxP-2* during embryogenesis may also be expressed in pluripotent neoblasts. In contrast, lineage-primed progenitor factions within the σ subclass probably arise during S5. Genes with functions in tissue-specific differentiation programs, including *soxB1* (*Lapan and Reddien, 2012*; *Monjo and Romero, 2015*), *pou2-3* (*Scimone et al., 2011*), *nos* (*Wang et al., 2007*), *pitx* (*Currie and Pearson, 2013*; *März et al., 2013*), *sim* (*Cowles et al., 2013*), and *smad6/7–1* (*González-Sastre et al., 2012*) were expressed

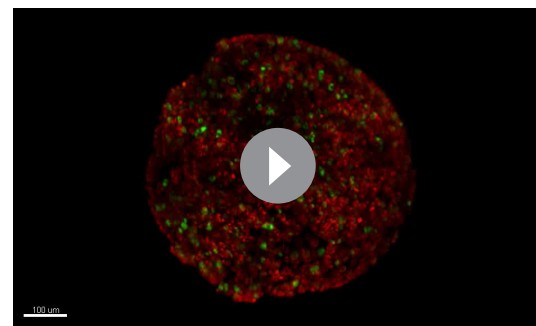

**Video 11.** Definitive epidermal progenitors arise in the *piwi-1+* blastomere population during S5. SPIM -reconstructed S5 embryo costained with *piwi-1* (red) and *p53* (green). Definitive epidermal progenitors, coexpressing *piwi-1* and *p53*, are dispersed in the embryonic wall. As definitive epidermal progenitors differentiate, they are predicted to downregulate *piwi-1* and to retain expression of *p53*.

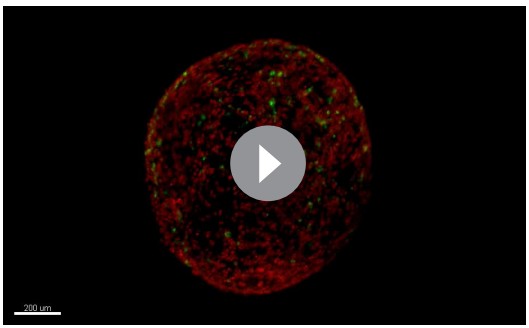

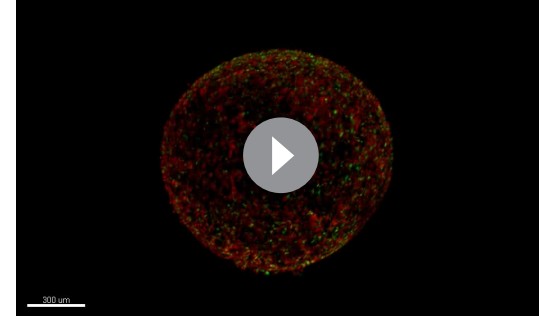

**Video 12.** Definitive gut progenitors arise in the *piwi-1+* blastomere population during S5. SPIM reconstructed S5 embryo costained with *piwi-1* and *VAL-like* (both in red) and *gata456a* (green). Definitive gut progenitors, coexpressing *piwi-1* and *gata456a,* are dispersed in the embryonic wall. As definitive gut progenitors differentiate, they are predicted to downregulate *piwi*-1 and to retain expression of *gata456a. VAL-like* is expressed the temporary embryonic pharynx and is not detected in *piwi-1+* blastomeres.

**Video 13.** Muscle progenitors arise in the *piwi-1+* blastomere population during S5. SPIM reconstructed S5 embryo costained with *piwi-1* and *VAL-like* (both in red) and *myoD* (green). Muscle progenitors, coexpressing *piwi-1* and *myoD*, are dispersed in the embryonic wall. As muscle progenitors differentiate, they are predicted to downregulate *piwi*-1 and to retain expression of *myoD. VAL-like* is expressed the temporary embryonic pharynx and is not detected in *piwi-1+* blastomeres.

at low levels in S2–S4 embryos and were upregulated during S5 (*Figure 6G*, *Figure 6—source data 1*, *Figure 1—source data 5*).

The gene expression signature of the adult neoblast compartment is an emergent property of the *piwi-1+* population during embryogenesis. First, EEE transcripts are uniquely associated with the expression signature(s) of undifferentiated *piwi-1+* blastomeres in early embryos. Second, adult neoblast subclasses arise as lineages are born within *piwi-1+* blastomeres during S5. Molecular heterogeneity within the neoblast compartment is largely attributed to the diverse array of lineage-dedicated progenitors within the population, a hypothesis supported by our finding that subclass marker expression was dramatically upregulated as organogenesis began. Progenitor subpopulations required for organ formation during embryogenesis persist into adulthood, where steady-state output from different lineages maintains tissue homeostasis. At present, we cannot distinguish whether lineages perpetually re-emerge due to asymmetric division of pluripotent stem cells, or whether progenitor populations established during embryogenesis are maintained by self-renewal. These observations beg the question: do *piwi-1+* cells behave similarly to neoblasts throughout embryogenesis? Or alternatively, is neoblast activity an acquired feature that emerges in tandem with the adult molecular expression signature?

## Embryos undergoing organogenesis contain cells with cNeoblast activity

Neoblasts are completely and irreversibly eliminated following treatment with 6,000 Rads (*Reddien et al., 2005*; *Wagner et al., 2011*), causing irradiation sickness and ultimately death. Transplantation of wildtype adult tissue grafts or

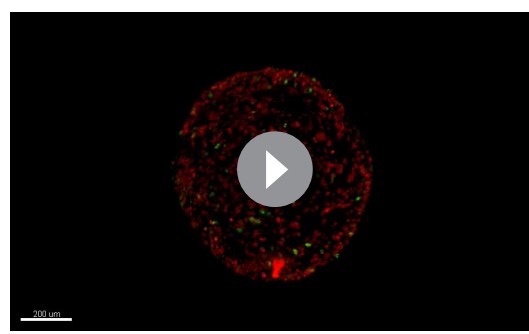

**Video 14.** Neural progenitors arise in the *piwi-1+* blastomere population during S5. SPIM reconstructed S5 embryo costained with *piwi-1* and *VAL-like* (both in red) and *pax6a* (green). Neural progenitors, coexpressing *piwi-1* and *pax6a*, are dispersed in the embryonic wall. As neural progenitors differentiate, they are predicted to downregulate *piwi*-1 and to retain expression of *pax6a. VAL-like* is expressed the temporary embryonic pharynx and is not detected in *piwi-1+* blastomeres.

cell suspensions into lethally irradiated adult hosts that are devoid of stem cells results in engraftment and expansion of donor-derived *piwi-1+* cells, production of differentiated progeny, reconstitution of the neoblast compartment and rescue from lethality (*Baguñà J and Auladell, 1989*; *Guedelhoefer and Sánchez Alvarado, 2012*; *van Wolfswinkel et al., 2014*; *Wagner et al., 2011*). *piwi-1+* cells that form pluripotent, expanding colonies following sublethal irradiation or transplantation into a lethally irradiated host are called clonogenic neoblasts (cNeoblasts) (*Wagner et al., 2011*). cNeoblasts are predicted to have a widespread distribution in the parenchyma, and this population contains within it the most primitive stem cells. At present, the operational definition for cNeoblast exists apart from the gene expression signatures for the neoblast subclasses; cNeoblasts are likely contained within, but may not be exclusive to, the σ class.

To assess whether *Smed* embryos harbor *piwi-1+* cells with cNeoblast activity, heterochronic, heterotopic transplantations were performed. S4, S5, S6, S7 and S8 embryonic cell suspensions were injected into the tail parenchyma of lethally irradiated sexual adult hosts at 3 days post-irradiation (dpi) (*Figure 7A*, Materials and Methods). To determine whether comparable numbers of *piwi-1+* cells were introduced per host for the developmental stages assayed, transplanted hosts were fixed at 1 hr post-transplant (hpt) and stained with *piwi-1* riboprobes. More than 85% of S4–S8 transplants fixed at 1 hpt contained *piwi-1+* cell(s) in the tail parenchyma, suggesting that the cell injection technique was robust and reliable (*Figure 7B*). S5, S6, S7 and S8 transplants contained comparable numbers of *piwi-1+* cells per host at 1 hpt, while significantly fewer *piwi-1+* cells were introduced per S4 embryonic cell transplant (*Figure 7C,F*).

To assay whether embryonic *piwi-1+* cells persisted and proliferated in an adult microenvironment, cohorts of transplanted animals and lethally irradiated uninjected hosts were fixed at 5 days post-transplantation (dpt) and stained with *piwi-1* riboprobes and H3S10p antibodies (*Figure 7B–F*). Persistent *piwi-1+* cells from S6, S7 and S8 embryos were observed in the vast majority of transplants scored at 5 dpt (*Figure 7B*), and no statistically significant difference among stages was detected in the mean number of *piwi-1+* cells present per host (*Figure 7C*). Moreover, most S6, S7 and S8 embryonic donor cell transplants contained dividing *piwi-1+* cell(s) at 5 dpt (*Figure 7D–F*), and no statistically significant difference among stages was detected in the mitotic index of donor-derived *piwi-1+* cells per host at 5 dpt (*Figure 7E*). In contrast, S4 and S5 embryonic *piwi-1+* cells were significantly less likely to persist in adult hosts at 5 dpt than were S6, S7 or S8 embryonic *piwi-1+* cells (*Figure 7B–C*). Persistent S4 derived *piwi-1+* cells were rarely observed at 5 dpt, and cell division was not observed (*Figure 7D–E*). As expected, *piwi-1+* cells were never observed in uninjected lethally irradiated hosts (n = 82 hosts scored at 3 dpi).

Fewer S5 embryonic cell transplants contained persistent *piwi-1+* cells at 5 dpt (*Figure 7B*), and the number of S5 derived *piwi-1+* cells per host at 5 dpt was significantly lower than for later stages (*Figure 7C*). Similarly, the fraction of S5 embryonic cell transplants containing mitotic *piwi-1+* cell(s) was reduced relative to S6 and later stages (*Figure 7D*). The reduced persistence of S4 and S5 derived *piwi-1+* cells in an adult microenvironment was probably not attributable to technical variability or to the absolute number of embryonic *piwi-1+* cells introduced per host, since comparable numbers of S5, S6, S7 and S8 *piwi-1+* cells were introduced per transplant (*Figure 7C*). Stage-specific, cell autonomous factors likely underlie the starkly different responses of S5 and S6 embryonic cells following transplantation into the adult parenchyma. These results suggest that S4 and S5 *piwi-1+* blastomeres are functionally distinct from *piwi-1+* cells present at S6 and later stages. *piwi-1+* blastomeres acquire competency to engraft and respond appropriately to adult environmental cues as development proceeds during S5.

To assess whether embryos undergoing organogenesis harbor cells capable of rescuing and reconstituting lethally irradiated adult hosts, S5, S6, S7 and S8 embryonic cell suspensions were injected into lethally irradiated sexual adult hosts at 1 dpi (*Figure 8A*, Materials and methods). Consistent with previous results, persistent *piwi-1+* cells from S6, S7 and S8 embryos were observed in the vast majority of samples scored at 5 dpt (*Figure 8B*), and no statistically significant difference was detected in either the mean number or the mitotic index of *piwi-1+* donor-derived cells per host at 5 dpt (*Figure 8C–D*). Likewise, donor-derived *piwi-1+* cells from S5 embryos were far less likely to persist and divide in adult hosts (*Figure 8B–D*). As expected, *piwi-1+* cells were never observed in uninjected lethally irradiated hosts (n = 81 hosts scored at 6 dpi).

Survival of S5–S8 transplants and host controls was monitored for 70 dpt (*i.e.,* 71 dpi). Worms were scored for irradiation-induced phenotypes, including head and tail regression, lesion formation,

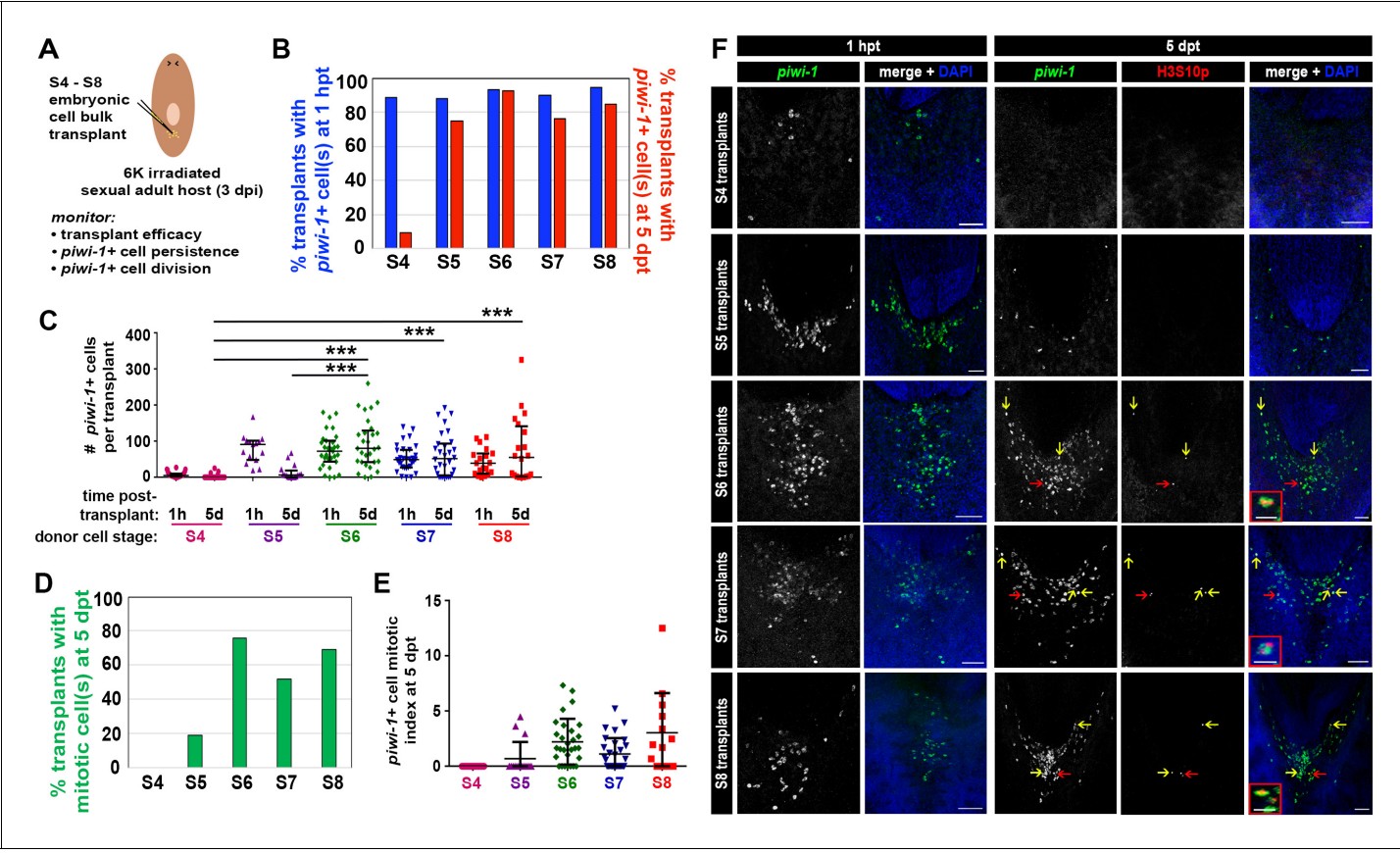

**Figure 7.** Embryonic cells acquire the ability to engraft, persist and proliferate in an adult microenvironment as organogenesis proceeds. (**A**) Schematic depicting the workflow for heterochronic transplantation experiments. S4, S5, S6, S7 or S8 embryonic cell suspensions were injected into the tails of lethally irradiated sexual adult hosts at 3 days post-irradiation (dpi). Cohorts of transplanted animals were fixed at 1 hr and 5 days post-transplantation (1 hpt and 5 dpt, respectively) for staining with *piwi-1* riboprobes and H3S10p antibodies. Lethally irradiated, uninjected host controls were fixed and stained at 5 dpt. (**B**) Percentage of transplanted animals fixed at 1 hpt (blue bars) or 5 dpt (red bars) containing one or more donor-derived *piwi-1+* cell (s). X-axis: stage (S) of donor cells. (**C**) Number of donor-derived *piwi-1+* cell(s) per transplant at 1 hpt and 5 dpt. Each point represents one transplanted animal. Mean ± standard deviation (black bars) are shown. Statistical tests were performed using a generalized linear model, assuming that the counts followed a Poisson distribution. S4 transplants contained significantly fewer *piwi-1+* cells at 1 hpt than S5, S6, S7 or S8 transplants (Tukey post-hoc comparisons, S4 vs S5, S4 vs S6 and S4 vs S7, S4 vs S8: p<0.001). Group differences in the number of *piwi-1+* cells at 1 hpt for S5 and S6 transplants were not statistically significant (p=0.21). Significantly fewer S4 and S5 donor-derived *piwi-1+* cells persisted at 5 dpt than were observed for later stages (Tukey post-hoc comparisons: S4 vs S5, S4 vs S6, S4 vs S7 and S4 vs S8: p<0.001. S5 vs S6, S5 vs S7, S5 vs S8: p<0.001). (**D**) Percentage of transplants with mitotic *piwi-1+* cell(s) at 5 dpt (green bars). X-axis: Donor cell stage. (**E**) Mitotic index for donor-derived *piwi-1+* cells at 5 dpt. Stage-specific differences were not observed for S4–S8 embryonic cell populations using a generalized linear model, assuming counts followed a Poisson distribution and the number of *piwi-1+* cells as a covariate. (**B–E**) Numbers of transplants scored: S4: n = 36 (1 hpt), n = 43 (5 dpt), four independent experiments. S5: n = 15 (1 hpt), n = 16 (5 dpt), two independent experiments. S6: n = 31 (1 hpt), n = 29 (5 dpt), four independent experiments. S7: n = 31 (1 hpt), n = 30 (5 dpt), four independent experiments. S8: n = 19 (1 hpt), n = 20 (5 dpt), three independent experiments. (**F**) Confocal maximal projections of S4, S5, S6, S7 and S8 embryonic cell transplants fixed at 1 hpt and 5 dpt. Animals were stained with *piwi-1* riboprobes (green), antibodies against the mitotic marker H3S10p (red, 5 dpt only) and DAPI nuclear counterstain (blue). S6, S7 and S8 insets: mitotic *piwi-1+* cells. Red arrows indicate mitotic cells magnified in insets. Yellow arrows: mitotic *piwi-1+* cells. Scale bar (inset): 20 µm. Scale bar (panel): 100 µm. (**B–C**) S4–S8 embryonic *piwi-1+* cells were reliably introduced into hosts. S6–S8 embryonic *piwi-1+* cells persisted in an adult microenvironment. (**D-E**) S6–S8 embryonic *piwi-1+* cells proliferated in an adult microenvironment.

ventral curling and death. Rescue, anterior blastema formation and subsequent regeneration of an entire individual from injected tail fragment, was also scored (*Guedelhoefer and Sánchez Alvarado, 2012*; *Wagner et al., 2011*) (*Figure 8F–G*, *Figure 8—figure supplement 1*). Irradiation-induced phenotypes manifest in control and transplanted individuals alike between 7–14 dpt (*i.e.*, 8–15 dpi) (*Figure 8—figure supplement 1A–E*), and none of the controls survived long-term (*Figure 8F*, *Figure 8—figure supplement 1E*). Remarkably S6, S7 and S8 embryonic donor cells were capable of

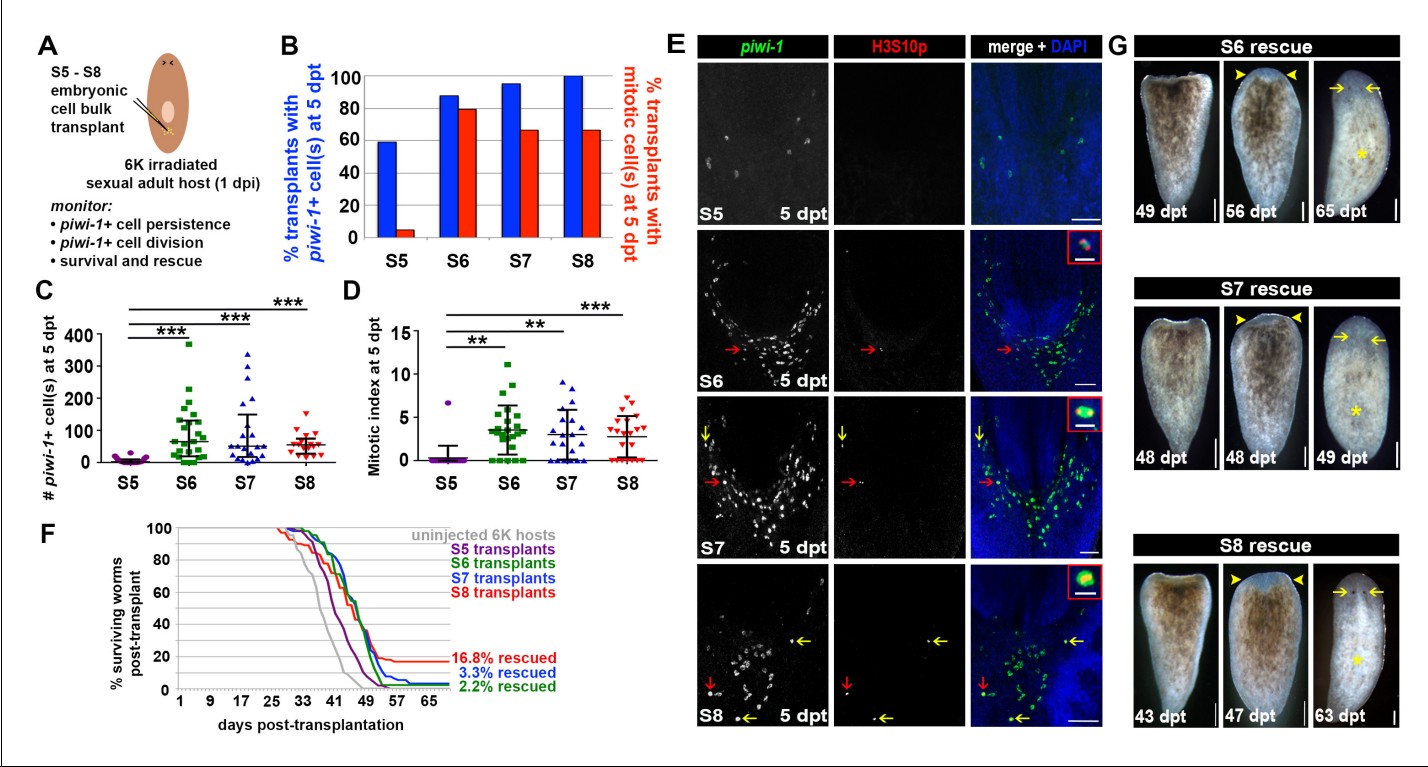

**Figure 8.** Embryos undergoing organogenesis contain cNeoblasts. (**A**) Schematic for heterochronic transplantation experiments. S5, S6, S7 or S8 embryonic cell suspensions were injected into the tail parenchyma of lethally irradiated sexual adult hosts at 1 day post irradiation (dpi). Cohorts of transplanted animals and uninjected host controls were fixed at 5 days post-transplantation (dpt) for staining with *piwi-1* riboprobes and H3S10p antibodies. The remaining animals were monitored for 70 dpt for survival and rescue. (**B**) Percentage of transplants with persistent, donor-derived *piwi-1* + cell(s) (blue) or donor-derived mitotic (*piwi-1+*, H3S10p+) cell(s) (red) at 5 dpt. X-axis: Embryonic donor cell stage. (**C**) Number of *piwi-1+* cells per transplanted host at 5 dpt for S5–S8 embryonic cell transplants. Each point represents one transplanted animal. Means ± standard deviation (SD) are shown (black bars). Statistically significant differences in the number of persistent *piwi-1+* cells per transplant at 5 dpt were observed using a generalized linear model, assuming that count data followed a Poisson distribution. S5 transplants contained fewer persistent *piwi-1+* cells than S6 or S7 transplants (Tukey post-hoc comparisons, S5 vs S6: p<0.0001, S5 vs S7: p<0.0001, S5 vs S8: p<0.0001). (**D**) Mitotic index for donor-derived *piwi-1+* cells at 5 dpt for S5–S8 embryonic cell transplants. Each point represents one transplanted animal. Means ± standard deviation (SD) are shown (black bars). Statistically significant differences in the *piwi-1+* cell mitotic index were observed using a generalized linear model with *piwi-1+* cell counts as a covariate, assuming that count data followed a Poisson distribution. S5 transplants contained significantly fewer cycling cells than S6, S7 or S8 transplants (Tukey post-hoc comparisons, S5 vs S6: p<0.01, S5 vs S7: p<0.01, S5 vs S8: p<0.001). (**E**) Confocal maximal projections for S5, S6, S7 and S8 embryonic cell transplants fixed at 5 dpt and stained with *piwi-1* riboprobes (green), H3S10p antibodies (red) and DAPI (blue). S6, S7 and S8 insets show mitotic *piwi-1+* cells. Red arrows indicate mitotic cells magnified in the insets. Yellow arrows: mitotic *piwi-1+* cells. Scale bar (inset): 20 µm. Scale bar (panel): 100 µm. (**B–E**) Numbers of transplants scored in four independent experiments: S5 n = 22; S6 n = 24; S7 n = 21; S8 n = 27 in (**C**), n = 21 in (**D**). (**F**) Survival curves for S5, S6, S7 and S8 embryonic cell transplants and uninjected 6,000-Rad-irradiated host controls as a function of time (days) post-transplant. (**G**) Live images of regenerating S6, S7 and S8 rescue animals. Left: Tail fragment after self-amputation of head and trunk tissue. Middle: Tail fragment with unpigmented anterior blastema (yellow arrowheads). Right: Animal with new head tissue and developing eyes (yellow arrows) and a regenerated pharynx (yellow asterisk). Animals from different experiments are shown in the S7 panels; the same animals are shown in the S6 and S8 panels. Dorsal view. Anterior: top. Scale: 100 µm. (**F–G**) Numbers of transplants scored in four independent experiments: host controls n = 89; S5 n = 105; S6 n = 90; S7 n = 92; S8 n = 85. Rescue animals were obtained in two experiments for S6 and S7 transplants, and four experiments for S8 transplants. (**B–E**) S6, S7 and S8 embryonic donor cells persist and divide in the adult parenchyma. (**F–G**) S6, S7 and S8 embryonic cells can rescue lethally irradiated adult hosts.

The following figure supplement is available for figure 8:

**Figure supplement 1.** Progression of irradiation-induced phenotypes, rescue or death for heterochronic transplantation assays.

rescuing lethally irradiated adult hosts (*Figure 8F–G*). Rescued animals underwent complete head regression, self-amputation posterior to the pharynx, and tail fragments that formed anterior blastemas which promoted regeneration of individuals containing two visible eyes, a central pharynx and triclad gut (*Figure 8F–G*, *Figure 8—figure supplement 1B–D*). None of the S5 embryonic cell recipients mounted a rescue response (*Figure 8F*, *Figure 8—figure supplement 1A*). The rescue assay results suggest that S6, S7 and S8 embryos harbor cNeoblasts, cells that are capable of self-renewing and producing the diverse array of cell types required for whole animal regeneration.

Taken together, stage-dependent molecular and functional distinctions exist among *piwi-1+* blastomeres before and after organogenesis begins. S4 *piwi-1+* cells, which express EEE and adult asexual neoblast enriched transcripts, are largely incapable of persisting and dividing in an adult microenvironment (*Figure 7B–C*). During S5, as dramatic shifts in gene expression occur, *piwi-1+* cells become competent to persist and proliferate in the adult parenchyma (*Figures 7B–C* and *8B–D*). We propose that cNeoblasts arise during S5. Heterochronic transplantation experiments revealed that S6, S7 and S8 embryos possess cells that behaved similarly to adult cNeoblasts: they consistently engrafted into adult hosts, proliferated and were ultimately capable of mounting a rescue response (*Figure 8B–G*). Acquisition of cNeoblast activity during embryogenesis correlates temporally with large-scale changes in gene expression observed at the outset of organogenesis, suggesting that pluripotent stem cells and lineage-primed progenitors first emerge during S5 (*Figure 9*). The ontogeny of the adult neoblast compartment can therefore be traced back to the *piwi-*

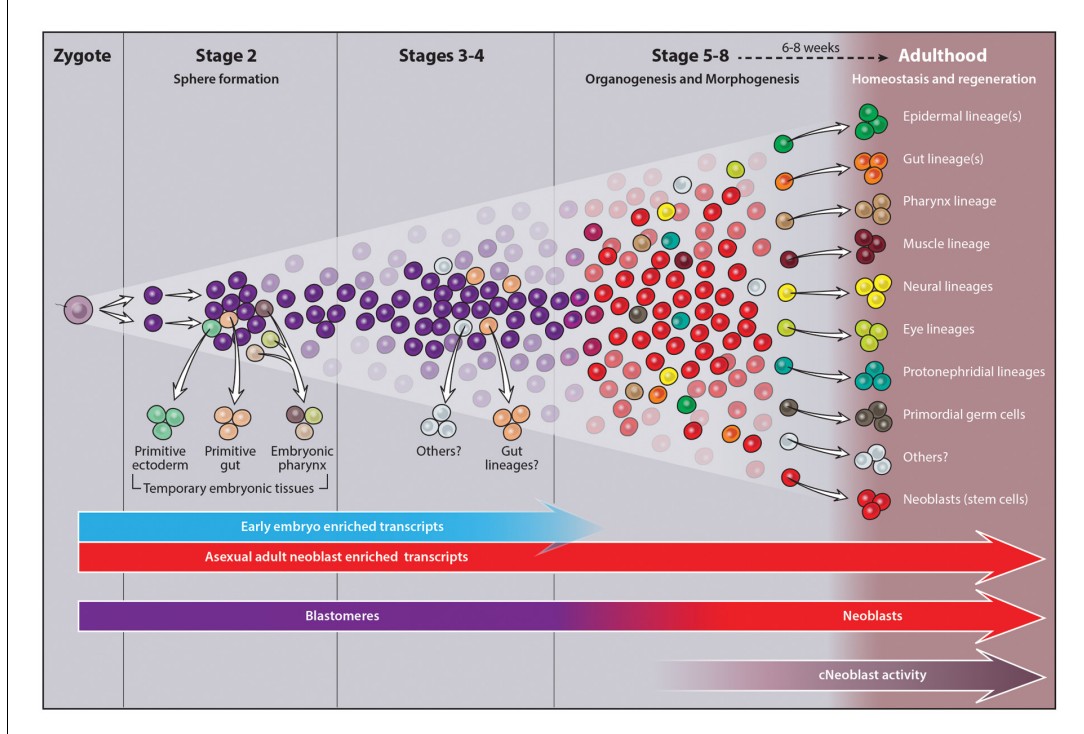

**Figure 9.** Ontogeny of the adult neoblast compartment. Asynchronously cycling *piwi-1+* cells fuel embryogenesis, giving rise to all temporary and definitive tissues. During S2, some *piwi-1+* blastomeres (purple cells) exit the cell cycle and differentiate into temporary embryonic tissues (primitive ectoderm, temporary embryonic pharynx and primitive endoderm). The remaining *piwi-1+* blastomeres, located in the embryonic wall (purple cells, S3-S4), continue to divide and express both EEE transcripts (turquoise arrow) and adult asexual neoblast enriched transcripts (red arrow). As organogenesis begins during S5, EEE transcripts are downregulated throughout the compartment (purple cells transition into red). Concomitantly, progenitor subpopulations required for definitive organ formation are specified via the heterogeneous expression of developmental transcription factors within the *piwi-1+* population (colored cells denote different progenitor subpopulations). Adult pluripotent neoblasts, themselves a lineage, are established during S5 (red cells). Embryonic donor cells harvested during or after S6 function similarly to adult neoblasts (cNeoblast activity, gray arrow). Pluripotent and lineage-primed neoblasts established during embryogenesis are maintained throughout the lifetime of the animal. Neoblasts are required for tissue maintenance during homeostasis and the formation of new tissue during regeneration.

*1+* zygote, which gives rise to anarchic, cycling *piwi-1+* blastomeres, some of which persist in the S3–S4 embryonic wall and establish pluripotent neoblasts and progenitor subpopulations during S5. The remarkable developmental plasticity of adult planarians is likely due to the singular ability of neoblasts to perpetuate and redeploy embryonic developmental programs.

## Discussion

Regeneration remains one of the most poorly understood processes in developmental biology. The origin and regulation of cells that make regeneration possible are largely obscure. How similar or distinct developmental processes are during embryogenesis and regeneration remains to be determined. In fact, few experimental systems are available to study these issues systematically. Here, we lay the foundation for the formal comparison of molecular processes and gene functions during embryogenesis and regeneration in *Smed*, an organism uniquely suited to address the relationship between developmental plasticity and regeneration competency.

### *S. mediterranea*: a developmental system for comparative studies of embryogenesis and regeneration

To identify similarities and key differences between embryogenesis and regeneration, careful consideration must be given to distinctions in context, chronology, scope and type of regeneration (homeostatic or facultative), and to the ontogeny of effector cell type(s). Which aspects of embryonic development are recapitulated during regeneration, and which are context-specific? What factors influence the competency and extent of regenerative responses, and do these factors have embryonic origins? The workhorses of modern developmental biology, *C. elegans*, *D. melanogaster*, *D. rerio*, and *M. musculus* have limited, if any, regenerative potential during adulthood, precluding or severely limiting comparative inquiries. In contrast, *Smed* adults exhibit robust homeostatic and facultative regenerative potential, both neoblast-dependent phenomena that have largely been studied using a clonal, asexually reproducing strain. Neither descriptive nor functional studies of *Smed* embryogenesis have been reported.

We generated two foundational resources: a molecular staging series for *Smed* embryogenesis and an expression atlas describing temporary and definitive organ development. To investigate the embryonic origins of regeneration, we provide an ontogeny for the adult neoblast compartment. The molecular staging series facilitated the identification of embryonic predecessors of adult neoblast lineages and the developmental transition when neoblast specification occurs. Adult neoblast lineages, including pluripotent stem cells and lineage-primed progenitors, are established as definitive organogenesis begins. Neoblast lineages, required for organ formation during embryogenesis, persist into adulthood, where they are redeployed for homeostatic maintenance of all differentiated tissues, including the germline, and formation of missing tissues during regeneration. Moving forward, investigation of neoblast dynamics, particularly the regulation of self-renewal and commitment to differentiation, will be a central, unifying feature of comparative studies on organ formation, maintenance, repair and replacement.

### Deciphering the origin and identity of neoblasts, agents of development, tissue homeostasis and regeneration

Unraveling the molecular mechanism of neoblast specification during embryogenesis will provide insight into the identity of adult pluripotent stem cells indispensible for homeostatic and facultative regeneration in planarians. Intensive efforts to define the term neoblast, and to characterize the presumed plurality of cell types within the asexual adult neoblast population, have generated numerous, unreconciled molecular and functional criteria. At present, gene expression signatures for neoblast subpopulations cannot be correlated with functional distinctions (should they exist) in self-renewal, pluripotency or cell fate restriction. It is not known whether hierarchical relationships exist between pluripotent cells and progenitors, whether cycling progenitors can dedifferentiate and/or interconvert, how stable subpopulations are over time, or when and how progenitors make an irreversible commitment to differentiation. Furthermore, preferred usage of *Smed* C4 asexual animals to classify and examine relationships among subpopulations precludes comprehensive examination of the lineage repertoire.

The expression of pluripotency factors is probably necessary, but not sufficient, for the assumption of neoblast fate. We report that many genes known to regulate adult asexual neoblast self-renewal and maintenance were expressed throughout embryogenesis, and showed that several of these genes were expressed throughout S3–S5 piwi-1+ blastomeres. Functional differences between blastomeres and neoblasts were observed in heterochronic, heterotopic transplantation experiments. S4 and S5 embryonic donor cells were far less likely to persist and proliferate in stem cell deficient adult hosts than were embryonic donor cells harvested during S6 or later. Moreover, S5 embryonic donor cells did not rescue lethally irradiated hosts, whereas S6–S8 embryonic donor cells functioned like adult neoblasts and were capable of rescue. These results suggest that neoblast specification occurs during S5, and highlight the importance of cell-intrinsic changes in the blastomere to neoblast transition.

Differences in transplanted cell behavior correlated with large-scale changes in gene expression as definitive organogenesis began. EEE transcripts, uniquely expressed in blastomeres, are attractive candidates for repressors of neoblast fate, and/or effectors of molecular changes in blastomeres that are necessary for acquisition of neoblast fate. It is not known whether EEE transcript downregulation in blastomeres is necessary for neoblast specification, or whether expression of EEE transcripts and developmental TFs required for lineage specification are mutually exclusive. Further studies will determine whether and how maternal factors influence the early stages of Smed embryogenesis. Which EEE transcripts are maternally deposited, and which mechanisms effect maternal transcript degradation? When do wave(s) of zygotic genome activation occur, and how does zygotic genome activation in blastomeres relate to cell differentiation and neoblast specification? Additional factors, such as changes in chromatin state, may also correlate with or play a causative role in the blastomere to neoblast transition.

## The paradox of pattern formation: robustness of an anarchic, self-organizing system and its implications for understanding regeneration

Ectolecithal development and dispersed cleavage pose unique developmental challenges, making Smed embryogenesis a novel paradigm for regulative development. Herein, the collective pluripotency of an anarchic, cycling piwi-1+ population generates the diversity of cell types required for the development of all temporary and definitive organ systems in these bilaterally symmetric, triploblastic animals. We showed that piwi-1+ blastomeres are spatially disordered within the embryonic wall during S3–S4, and that dispersed epidermal, gut, muscle and neural progenitors arise within the blastomere population during S5. Further investigation is needed to understand how signals from differentiated tissues impact cell fate decisions within the piwi-1+ blastomere compartment and effect progenitor cell migration, communication and interactions necessary to form organ rudiments.

Smed embryogenesis provides a unique vantage point from which to investigate the origin, anatomical composition and signaling logic underlying the neoblast niche. During sphere formation, differentiating blastomeres must interact and self-assemble temporary embryonic tissues, providing structure to the embryo and establishing a microenvironment that promotes maintenance and expansion of piwi-1+ blastomeres. In turn, blastomeres are likely to effect changes that are conducive to the establishment and/or maintenance of the neoblast population. Neoblast lineages are predicted to actively maintain their niche during the lifetime of the animal. Although the adult gastrovascular system has long been suspected of providing trophic support signals to neoblasts (*Forsthoefel et al., 2012*), the molecular mechanisms underlying this phenomenon remain elusive. Expansion of piwi-1+ blastomeres correlates with that of an ill-defined embryonic gut population (*Figure 1—figure supplement 13A–D*). Investigating gut development and gut communication with the piwi-1+ population during embryogenesis may uncover key regulators of neoblast specification or regulators of neoblast dynamics that may similarly impact stem cell behavior during adulthood.

We report that many transcripts implicated in lineage commitment and classification of neoblast subclasses wereexpressed at low levels in early embryos and were dramatically upregulated as organogenesis began. This observation is consistent with the hypothesis that neoblast heterogeneity is due to the presence of different subpopulations of cycling, lineage-primed progenitors within the compartment (*Reddien, 2013*). It also suggests that organ formation during embryogenesis probably utilizes many of the same genetic regulatory networks and transition states elucidated during adult homeostasis and regeneration. Development of techniques to interrogate gene function during embryogenesis will enable us to identify master regulators of organogenesis for different

tissues, and to address similarities and differences in their modes of action during embryogenesis and adulthood. Embryogenesis may also provide a vantage point for the identification of upstream activators for these developmental TFs, helping to address how diverse, dispersed patterns of gene activation arise in the blastomere and neoblast compartments.

The key distinction between embryogenesis and regeneration is the de novo formation of tissues in the former, and the presence of preexisting structures and signaling environments in the latter. Studies can now be performed to assess how tissues that are hypothesized to be instructive for regeneration are initially established during embryogenesis, and how the formation of these tissues relates to the acquisition of regenerative potential during development. For example, planarian body wall muscle is hypothesized to be required for the re-specification of axial identities during regeneration (*Witchley et al., 2013*). At present, technical limitations preclude tissue-specific knock-down experiments that would address requirements for polarity genes in muscle during regeneration. However, we can now address when and how the definitive axes are established during *Smed* embryogenesis, including the identification of tissues and signals that initially polarize embryos prior to the development of body wall muscle. Examining axis formation during embryogenesis, and comparing the process across different chronological and developmental contexts, may uncover roles for additional tissues and/or novel polarity regulators. Furthermore, we can address which developmental milestones and gene products are required to establish a state of regeneration competency in the embryo.

Sustained effort and continued investment in the adaptation and development of new technologies for the molecular interrogation of *Smed* embryogenesis will facilitate discoveries that may challenge long-held assumptions about developmental processes, including cell fate specification, pattern formation and adult stem cell regulation. Moreover, utilizing *Smed* for comparative studies of embryogenesis and regeneration presents an unprecedented opportunity for formal examination of the embryonic origins of regenerative potential.

## Materials and methods

### Planaria culture and husbandry

Sexually reproducing *S. mediterranea* (*Smed*) stocks were descendants of animals collected in Sardinia by Dr. Maria Pala in 1999. Animals from the clonally derived sexual strain S2F1L3F2 (*Wagner et al., 2011*) and from the asexual clonal strain CIW-4 (C4) (*Newmark and Sánchez Alvarado, 2002*) were propagated via successive rounds of amputation and regeneration. Animals were maintained in 1x Montjuic water at 20°C in the dark and fed homogenized beef liver as previously described in *Cebrià and Newmark (2005)*. Cultures subjected to intensive cutting and/or feeding regimens were supplemented with 100 µg/mL gentamicin sulfate (Gemini Bioproducts, #400–100P).

Egg capsules were collected daily from outbred cohorts of sexually mature adults cultured at low density (6–8 animals per 400 mL culture), and were stored in dated Petri dishes at 20°C in constant darkness until use. The collection date was considered 1 day post-egg capsule deposition. To maintain optimal fertility levels, sexually mature animals used for egg capsule collections were replaced every 3–4 months with either juveniles (6–8 weeks post hatching) or adult regenerates (6–8 weeks post cut).

### Single embryo RNA-Seq

Live embryos were dissected out of egg capsules in 1x Holfreter's buffer (3.5 g/L NaCl; 0.2 g/L NaHCO$_3$; 0.05 g/L KCl; 0.2 g/L MgSO$_4$; 0.1 g/L CaCl$_2$; 1.0 g/L dextrose, pH 7.0–7.5) for S2–S7 egg capsules, or 1x Montjuic water for S8 hatchlings. Yolk (Y) samples were obtained from 8 d egg capsules that contained neither spherical nor elongating embryos. Single embryos were imaged on a Leica M205 FA dissecting microscope, transferred into microfuge tubes containing 200 µl TRIzol reagent (Thermo Fisher, item #15596–018), homogenized by pipetting, and stored at −80°C. Single animal samples of intact C4 adults and virgin, sexually mature adults were homogenized in 1.0 mL TRIzol using an IKA Ultra Turrax T 25 Basic tissue disruptor prior to storage at −80°C. Total RNA extraction was performed in 1.0 mL TRIzol per sample according to the manufacturer's protocol, following the recommendations for working with small amounts of tissue. Pellets were resuspended in 25 µl nuclease free water, and 5 µl aliquots were reserved for quality control testing. Total RNA

concentration and integrity were determined using Agilent Bioanalyzer 2100 Expert Total RNA Nano or Pico chips (Agilent Technologies, items # 5067–1511 and 5067–1513). Total RNA samples were prepared for ten biological replicates per time point, and total RNA quality and yield were considered along with embryo size and morphology when selecting samples for library construction.

PolyA-selected, single-stranded RNA-Seq libraries were prepared for four biological replicates per stage using the Illumina TruSeq RNA Sample V2 kit (item # RS-122–2001 and RS-122–2002), starting with 500 ng total RNA per sample (C4, virgin sexual adult [SX], Y, S4, S5, S6, S7, S8), or 100 ng total RNA per sample (S2, S3). Library concentrations and insert sizes were determined using Agilent Bioanalyzer DNA 1000 chips (Agilent Technologies, item # 5067–1504), and diluted, pooled samples were reanalyzed with Agilent Bioanalyzer 2100 DNA High Sensitivity chips (Agilent Technologies, item # 5067–4626). Nine barcoded samples, one replicate per time point (S2–S8, Y, C4) were pooled and sequenced per flow cell lane. Single end, 50 bp reads were acquired on an Illumina Hi-Seq 2000 sequencer. Illumina Primary Analysis version RTA 1.13.48.0 and Secondary Analysis version CASAVA-1.8.2 were run to demultiplex reads and generate FASTQ files. Barcoded SX replicates were pooled and run on a single flow cell lane of a HiSeq 2500, and Illumina Primary Analysis version RTA 1.17.21.3 and Secondary Analysis version CASAVA-1.8.2 were used. The RNA-Seq data have been deposited in the GEO database under the accession number GSE82280.

## Mapping sequencing reads to the smed_20140614 transcriptome

Sequencing reads were mapped to the smed_20140614 reference transcriptome (n = 36,035 transcripts) (*Tu et al., 2015*), which contains sequencing data from de novo Trinity assemblies from the C4 and sexual biotypes, staged embryo collections, sorted cycling (X1) cells, and previously published sources (*Adler et al. [2014]*, [*Böser et al. [2013]*]; the Dresden transcript collection at Plan-Mine [http://planmine.mpi-cbg.de]). Transcripts were consolidated and reduced to a unique set using the CD-HIT program (*Fu et al., 2012*). smed_20140614 sequences may be downloaded from http://smedgd.stowers.org. Reads were mapped using the Bowtie algorithm, Version 1.0.0 (*Langmead et al., 2009*), allowing for two mismatches and up to five multi-matches (–best –strata -v 2 m 5). Read counts for transcripts were tabulated from SAM files using a custom script. Of 36,035 transcripts, 32,000 accumulated $\geq$1 CPM across all 40 samples. Samples were each sequenced to an average depth of 19 million reads, and exhibited an average map rate of 89% to the transcriptome.

## RPKM normalization

RPKM (Reads Per Kilobase per Million) values were scaled using TMM normalization (*Robinson et al., 2010*) in edgeR to account for read depth across samples. In addition, 16s ribosomal RNA transcripts (SMED30032887), which soak up a significant but variable fraction of reads per sample, were removed prior to calculating RPKM values.

## Identification of differentially expressed transcripts

Differential gene expression was evaluated using the edgeR library (*Robinson et al., 2010*), and adjusted p-values were calculated as described in *Hochberg (1995)*. Pairwise comparisons were performed between adjacent time points using edgeR: Y vs S2, S2 vs S3, S3 vs S4, S4 vs S5, S5 vs S6, S6 vs S7 and S7 vs S8. Mapped data were filtered to remove transcripts with less than a sum of 1 CPM across all 32 samples, resulting in 30,766 transcripts. The maximum read sum across samples for omitted transcripts was 14. In addition, transcripts for the 28S (SMED30027845), 18S (SMED30032663) and 16S (SMED30032887) ribosomal subunits were removed. Differentially expressed genes were identified in mixed stage reference comparisons using the GLM approach in edgeR to contrast each treatment group (i.e., developmental stage) to the average of the remaining groups (Y, S2–S8). Non-redundant lists of enriched transcripts from the pairwise and mixed stage reference comparisons, for S2 through S8, were subject to Euclidean distance clustering using scaled RPKM data in edgeR (*Figure 1C–I*, *Figure 1—figure supplements 4–10*, *Figure 1—source data 2–8*).

## GO analysis

Gene Ontology (GO) terms (*Gene Ontology Consortium, 2015*) were assigned to smed_20140614 transcripts on the basis of homologous PFAM domains (*Finn et al., 2014*) and significant Swiss-Prot

hits (E-value $\leq$ 0.001), (*UniProt Consortium, 2015*). GO term enrichment queries were performed using the R software package topGO, version 2.20.0 (*Alexa and Rahnenfuhrer, 2010*). GO analysis was performed on the non-redundant lists of enriched transcripts for S2–S8 (*Figure 1—source data 1*). Categories containing similar and/or related Biological Process (BP) GO ids enriched at one or more time point(s) were generated manually (*Figure 1—source data 1*). Enriched BP GO ids selected for categorization had Benjamini-Hochberg corrected p-values $\leq$1e-10 (*Benjamini and Hochberg, 1995*), and must have been associated with $\geq$1% of the enriched transcripts for the developmental stage(s) in question. BP GO ids were only assigned to one category. BP GO ids that did not describe a cell and/or tissue type present in *Smed* (e.g., heart, lung, neural crest) were omitted. Using these categories as a guide, lists of enriched BP GO ids and non-redundant lists of associated transcripts were generated for S2–S8 (*Figure 1—source data 2–8*). Transcripts may appear in more than one BP GO id category, just as transcripts may be associated with more than one GO term.

## Neoblast enriched transcript analysis

The neoblast enriched transcript list (n = 242) emerged from the downregulated sequences in whole animals at 24 and/or 48 hr post-lethal irradiation in three independent experiments (*Duncan et al., 2015*; *Wagner et al., 2012*) (Chen and Sánchez Alvarado, personal communication) (*Figure 4A*, *Figure 4—source data 1*). Euclidean distance clustering was performed using the mixed stage reference comparison data.

## Cloning

Constructs for riboprobe synthesis were constructed using the pPR-T4P (J. Rink) cloning strategy described in *Adler et al. (2014)*, with the exception that PCR inserts were amplified using mixed stage embryo cDNA (S2–S8) as a template. Primers used for cloning EEE transcripts and insert sequences appear in *Figure 5—source data 3*.

## Whole mount in situ hybridization (WISH) and immunostaining

Colorimetric and fluorescent WISH was performed as described by *King and Newmark (2013)* and *Pearson et al. (2009)*, with the following modifications:

1. 1) Egg capsules for S2–S7 embryos (2–10 days post-egg capsule deposition) were punctured with an insect pin and fixed in 4% formaldehyde in 1x PBS-Triton X (PBSTx) 0.5% for 4–6 hr at room temperature. Fixed embryos were dissected out from the egg capsules, washed in 1x PBSTx (0.5%) for 10 min, and subjected to incremental dehydration in methanol (10 min in 50% methanol, 2 × 10 min in 100% methanol). Fixed embryos were stored at −20°C. S2–S7 embryos were not bleached. Proteinase K treatment was increased to 20 min.
2. S8 embryos (newborn hatchlings, 14–16 days post-egg capsule deposition) were removed from egg capsules prior to fixation. S8 embryos and C4 intact adults (2–4 mm) were incubated in 4% formaldehyde in PBSTx (0.5%) for one hour at room temperature.
3. 3) S2–S7 embryos were not bleached. S8 embryos and C4 adults were bleached in formamide bleaching solution for 30 min to 1 hr under bright light.
4. Transplanted S2F1L3F2 hosts and irradiated intact controls (~5–6 mm in length) were fixed for 90 min. Worms were bleached for 1.5–2 hr in formamide bleaching solution under bright light.

Immunostaining was performed after fluorescent WISH development with rabbit polyclonal antibodies against H3S10p (1:1000; Millipore # 06–570), mouse monoclonal antibodies against Smith Antigen (Y12) (1:200, ThermoFisher Scientific, PIMA190490), or mouse monoclonal antibodies against *Smed* PIWI-1 (1:1000, a generous gift from J. Rink). H3S10p antibodies were detected using preabsorbed Alexa-conjugated secondary antibodies (1:1000, Abcam, ab150086, ab150069, ab150071), while anti-Y12 and anti-PIWI staining was visualized with tyramide development using Goat anti-mouse IgG F(ab')2 HRP (1:1000, Jackson Immunoresearch #115-036-072). Nuclear staining was performed with DAPI (1:5000, 1 mg/mL stock solution, ThermoFisher Scientific, D1306) or with Sytox Green (1:5000, 5 mM stock solution, ThermoFisher Scientific, S7020).

S2-–S8 colorimetric and fluorescent WISH samples that were to be imaged using light sheet microscopy were mounted as described in the Microscopy section, whereas others were mounted in

80% glycerol supplemented with 2.5% DABCO. C4 and sexual adult samples were mounted in Scale A2 mounting media (*Hama et al., 2011*).

## Histology

Embryos (S2–S8) were fixed for 4 hr at room temperature in 4% formaldehyde in 1x PBS, followed by 3 × 10 min washes in 1x PBS and gradual dehydration in 30%, 50%, 70%, 80%, 95% and 100% ethanol. The samples were soaked for 30 min in 5% glycerol diluted in 100% ethanol, cleared in xylene for 10 min, then soaked in two changes of Clear-rite 3 (Richard-Allan Scientific) for a total of 25 min. Paraffin infiltration proceeded with 2 × 45 min incubations, and embedded embryos underwent serial sectioning (5 µm thickness). Paraffin was removed prior to staining by heating slides at 60°C for 20 min, then performing 3 × 2 min washes in xylene, 3 × 1 min washes in 100% ethanol, 3 × 1 min washes in 80% ethanol before rinsing in tap water. Hematoxylin and eosin staining was performed using the ST Infinity H and E Staining System (Leica Biosystems) in a Leica Autostainer. Slides were incubated for 30 s in Hemalast, then for 2 min in hematoxylin, and were rinsed for 2 min in tap water. Next, slides were incubated for 45 s in differentiator and for 1 min in bluing agent, with each step followed by a 1 min tap water rinse and a 1 min incubation in 80% ethanol. Slides were stained with eosin for 30 s, dehydrated 3 × 1 min in 100% ethanol and cleared in 3 × 1 min incubations in xylene.

## Microscopy

A Leica M205 FA stereomicroscope was used to capture images of live animals and colorimetric WISH samples. A Leica DM600B upright microscope was used to capture images of histological sections. A Zeiss LSM-510-VIS confocal and a customized light sheet microscope were used to capture Z-stacks for fluorescent WISH samples.

Fixed, stained *Smed* embryos were mounted in 1% low melt agarose in 1x PBS along with fluorescent conjugated beads required for image registration and reconstruction (FluoSpheres Polystyrene Microspheres, 1.0 µm, red fluorescent [580/605], Invitrogen/Molecular Probes, F13083; FluoSpheres Carboxylate Modified Microspheres, 0.1 µm, yellow-green fluorescent [505/515], Invitrogen/Molecular Probes, F8803). 1 µM fluorescent bead stock solutions were diluted 1:10,000–1: 360,000, depending on the size of the embryo and the magnification of the detection objective used. Samples were placed in an imaging chamber within a Single Plane Illumination Microscopy (SPIM) system described in *Nakajima et al. (2013)*. S3–S5 embryos were imaged using either a 10x Plan Apochromat or a 5x Plan NeoFluar objective. Z-stacks were taken every 45° around the surface of the samples using a rotating stage, producing eight stacks of images per embryo. Multiview data sets were reconstructed using Fiji SPIM plugins for data registration and fusion (*Preibisch et al., 2010*). Reconstructed data sets were viewed in the Imaris software package, where they were cropped and masked to remove beads. Cell positions and the embryonic pharynx were marked manually using the 3D Spot Finder function, and the three-dimensional coordinates for marked cells were exported into excel for analysis of cell positions. Colocalization was determined manually on S3–S4 whole embryos, or on crop3D sections (100 µm X 200 µm X 100 µm) of S5 embryos.

### Cell position analysis on reconstructed SPIM images

Three-dimensional coordinates for *piwi-1+* cells or mitotic cells (*piwi-1+*, H3S10p+), the embryonic pharynx, and the embryo center were exported from SPIM reconstructions of S3 and S4 embryos. Spot positions collected in IMARIS were shifted and rotated in MATLAB to a coordinate system where the embryonic center was at the origin and the embryonic pharynx on the z-axis. Relative theta distribution likelihoods of the form $(1-\exp(-\theta/\theta'))*\sin(\theta)$ were calculated by assuming a $\theta'$ and calculating the likelihood that such a distribution would produce the observed data by multiplying together the individual probabilities. The uncertainty in the best fit $\theta''$ was found by simulating several datasets, each with a number of cells equal to those observed in the actual data and having a dampening term of $\theta''$. For each dataset, the most likely $\theta'$ was found, the standard deviation of which was taken to be the error in $\theta''$.

## Nanostring nCounter digital expression analysis

Total RNA samples from single embryos and adults (S2–S8, Y, C4 and SX; four biological replicates per sample) were prepared as described for single-animal RNA-Seq. 100 ng total RNA per sample was assayed on the Nanostring nCounter platform (*Geiss et al., 2008*) using a custom-made probe set. Reporter and capture probe sequences can be found in *Figure 5—source data 2*. Housekeeping control genes were reported in *Wenemoser et al. (2012)*. Nanostring data was normalized using the NanoStringNorm library from Bioconductor (*Waggott et al., 2012*), using the sum of the positive controls and the sum of the housekeeping genes as independent normalization factors, with the mean of the negative controls used to estimate background. Raw and normalized nanostring data can be found in *Figure 5—source data 2*.

## Bulk cell transplantation

Host animals (5–7 mm in length, ≥7 day starved) were selected from the clonally derived *Smed* sexual strain S2F1L3F2 (*Wagner et al., 2011*) and were cultured in 1x Montjuic water supplemented with 100 µg/mL gentamicin sulfate. The host neoblast population was ablated by exposure to 6,000 Rads on a GammaCell 40 Exactor irradiator; cohorts of unirradiated animals were reserved to verify complete elimination of the neoblast population by WISH with riboprobes against *piwi-1.*

Whole-embryo cell suspensions were created for S4, S5, S6, S7 or S8 by mechanically disrupting embryos in chilled, freshly made 1x Holfreter's solution +5% heat-inactivated fetal bovine serum (Sigma Aldrich, F4135) via repeated pipetting. Ten embryos were disrupted per cell suspension for S5, S6, S7 and S8, whereas 20–35 embryos were disrupted per S4 cell suspension. S4, S5 and S6 embryos were typically 'eviscerated' prior to pipetting by poking with an insect pin and gently squeezing out ingested yolk from the gut. Cell suspension volumes were adjusted to 1.0 mL before samples were filtered through 20 µm (all stages) and 10 µm (S5–S8) cell strainers (Partec CellTrics, Sysmex, 04-0042-2315 [20 µm], 04-0042-2314 [10 µm]) into low-retention microfuge tubes. Cells were pelleted by centrifugation at 310 rcf for 5 min, were resuspended in a final volume of ~10 µL (S5–S8) or ~5 µl (S4), and were kept on ice during transplantation.

Embryo cell suspensions were loaded by mouth pipetting into borosilicate glass needles (Sutter Instrument Co., #B100-75-15) pulled using a flaming/brown micropipette puller (Sutter Instrument Co., Model P-97) and were injected using an Eppendorf FemtoJet at 1.0–1.5 psi, as described by *Wagner et al. (2011)*. Hosts were immobilized on a cold peltier plate, ventral side up, and cells were injected into the tail stripe (i.e., the medial, post-pharyngeal parenchymal space between the two posterior branches of the intestine). Hosts were injected at 1 day post-irradiation (dpi) for rescue experiments and at 3 dpi for short-term experiments. Transplanted animals and uninjected, 6,000-Rad-irradiated hosts for rescue experiments were maintained individually in 3 cm petri dishes at 20°C in the dark, with water exchanges and visual inspection of animals performed every 2–3 days. Transplanted animals slated for fixation were reared in 10 cm petri dishes, with 10 or fewer animals per dish, with water exchanges every 2–3 days.

## Online data repository

Original data underlying this manuscript can be accessed from the Stowers Original Data Repository at http://www.stowers.org/research/publications/libpb-1086

## Acknowledgements

We thank members of the SIMR Planaria Core Facility MS Merryman, D Baumann, C Abrams, K Evans, J Thaden, J Theis and A Vogelslang for their invaluable assistance with planarian husbandry. N Marsh, N Thomas and S Beckham assisted with histology. K Zapian and J Jenkin assisted with irradiation for transplantation experiments. H Vu, S Elliott, LC Cheng, and K Tu graciously donated constructs; S Jones and C Guerrero assisted with WISH and transplant monitoring, respectively. K Tu and S Nowotarksi provided insightful comments on the manuscript. M Miller assisted with scientific illustration. H Li assisted with statistical analysis of the transplant experiments. Funding from the Stowers Institute for Medical Research, the Howard Hughes Medical Institute, and NIH RO1 R37GM057260-17 to ASA supported this work.

# Additional information

## Competing interests

ASA: Reviewing editor, *eLife*. The other authors declare that no competing interests exist.

## Funding

| Funder | Grant reference number | Author |
| --- | --- | --- |
| Howard Hughes Medical Institute | | Kai Lei |
| National Institute of General Medical Sciences | R37GM057260-17 | Alejandro Sánchez Alvarado |

The funders had no role in study design, data collection and interpretation, or the decision to submit the work for publication.

## Author contributions

ELD, Conceptualization, Investigation, Resources, Writing, Original draft, Writing, Review and editing, Visualization, Project administration; KL, Investigation; CWS, EJR, KG, Software, Formal analysis; AEK, Investigation, Visualization; SAM, Formal analysis, Visualization; LG, Investigation, Resources; SMCR, Software; ASA, Conceptualization, Resources, Writing, Review and editing, Supervision and funding acquisition

## Author ORCIDs

Alejandro Sánchez Alvarado, http://orcid.org/0000-0002-1966-6959

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
