## [Decision Letter]

Thank you for submitting your article "Embryonic origin of adult stem cells required for tissue homeostasis and regeneration" for consideration by *eLife*. Your article has been reviewed by four peer reviewers, and the evaluation has been overseen by Utpal Banerjee as Reviewing Editor and Marianne Bronner as the Senior Editor. The following individuals involved in review of your submission have agreed to reveal their identity: Diana Laird (Reviewer #2); Peter W Reddien (Reviewer #3).

The reviewers have discussed the reviews with one another and the Reviewing Editor has drafted this decision to help you prepare a revised submission. All reviewers agree on the importance of the work and the changes asked for can be addressed by making changes to the manuscript.

Summary:

In this manuscript, Davies and colleagues investigate the developmental origins of adult pluripotent stem cells (neoblasts) in the planarian *Schmidtea mediterranea*. The authors provide a comprehensive characterization of *S. mediterranea* embryogenesis by meticulously cataloguing (with corresponding gene expression data and expression mapping) the different stages of embryonic development. This groundwork allowed the authors to ask when neoblasts originate during embryogenesis. The authors demonstrate that neoblasts arise from anarchic, cycling *piwi*-1+ cells, which generate all temporary and definitive tissues/organs during embryonic development. Interestingly, early embryonic *piwi*-1+ cells have a distinct molecular signature from neoblasts. Unlike neoblasts, which express transcription factors (TFs) important for development, early embryonic *piwi*-1+ cells express "early embryo enriched" (EEE) transcripts. Later in embryogenesis when organogenesis initiates, EEE transcripts are downregulated while various developmental TFs are upregulated. Strikingly, in an elegant set of transplantation experiments, the authors demonstrate that *piwi*-1+ cells are functionally different at different embryonic stages. Similar to transplanted adult neoblasts, transplanted *piwi*-1+ cells from embryos undergoing organogenesis can rescue lethally irradiated planarians that are devoid of neoblasts, whereas, early EEEhi/TFlo *piwi*-1+ cells cannot.

Essential revisions:

Usually for minor revisions, we make a summary of the reviewers' comments, but in this case, the majority of questions raised are regarding lack of clarity in the manuscript and the complexity of data presentation. So we are including the detailed comments below. All of these can be addressed by rewriting the manuscript making it more accessible to readers. Also, below I paste an exchange between the reviewers during the post review session:

" I did want to address one point raised by two other reviewers, which I think reflects the way in which the data presentation could be improved. Both raised concerns about the transplant experiments that were easy to miss because some of the key results were buried in the supplement.

The "no transplant" control is shown in Figure 7. As expected (and shown previously) no planarians survive irradiation. Figure 7—figure supplement 2 shows *piwi*-1+ cell counts. In panel B the authors show the percentage of transplants with *piwi*-1+ cells at 1 hpt (directly after transplantation), which indicates that "the cell injection technique was robust and reliable". Panel C shows the number of *piwi*-1+ cells per transplant at 1 hpt. For example, in Figure 7—figure supplement 2, one can see that comparable numbers of *piwi*-1+ cells were transplanted from S5 and S6 embryos (1 hpt) but at 5 dpt, very few to zero *piwi*-1+ cells remain from S5 donors compared to S6 (panel E). Furthermore, irradiated hosts transplanted with S5 *piwi*-1+ donor cells are not rescued (Figure 7). The authors mention these data in the text (subsection “Embryos undergoing organogenesis contain cells with cNeoblast activity”), but it might be helpful to the reader if these data were in the main figure and not buried in the supplement. Furthermore, some of the panels can be merged (for example the 1 hpt and 5 dpt *piwi*-1+ cell transplants) so that it is easier for the reader to compare the data. "

As you can see, the reviewers, as much as they admired the science, were confused by the presentation. So, please make the paper a bit more lucid for the readership of *eLife*.

Reviewer #1

This impressive body of work will be a valuable resource to planarian biologists. To make the manuscript accessible to a broader audience and more manageable for readers in general, here are some editing suggestions that should be addressed before publication:

1) To appeal to the broader *eLife* audience, the authors should provide more description/context in the main text to help non-experts understand the results presented. This includes data the authors deem not essential for the narrative of the paper and is found in Supplementary Online Material instead of the main text. For example, sphere formation (subsection “Anarchic, cycling *piwi*-1+ blastomeres fuel Smed embryonic development”) is not defined in the main text, but is described in the SOM as beginning during Stage 2. Furthermore, the distinction between temporary and definitive embryonic organs may not be entirely clear to non-expert readers. The authors do a better job describing this uncommon developmental phenomenon in the SOM (e.g., "The embryonic pharynx degenerates during S6, as the definitive pharynx primordium develops beneath it. Embryonic pharynx markers are no longer detectable by S7").

2) Throughout the manuscript, the authors are urged to present figures in the order they are referred to in the text. Looking up figures (and multiple Supplements) that are "out of order" is cumbersome, especially when considering the sheer volume of data the reader must go through. For example, Figure 1—figure supplement 12 is referred to twice in the text (Results section) before any of the other Figure 1 supplements. Figure 1—figure supplement 12 should be designated as Figure 1—figure supplement 1 (and all other Figure 1 supplements renumbered accordingly thereafter). Also, having two sets of differently named supplemental data (Figure X—figure supplement Y, which is per the journal's style, and Supplementary Online Figure Z, which appears not to be) makes navigating supplementary material unnecessarily difficult for the reader.

3) Introduction section: Remove the extensive discussion of results from the Introduction. This will also help shorten the overall length of the paper.

Reviewer #2

Davies et all provide the first comprehensive description of *Smed* embryos at a molecular and anatomic level. This is not trivial given that *S. med* broods its embryos and the authors should be congratulated for this Herculean task. They specifically ask whether neoblasts that have been well studied in the adult planarian exist during embryogenesis and conclude that a cycling *piwi1*+ precursor population gives rise to organs as well as neoblasts, but is distinct on a molecular level as well as functionally in transplantation assays. Although important for the field, the following issues should be addressed:

Comments:

1) Overall, the accessibility and readability for a generalist journal as this one should be increased. This can be accomplished by presuming less prior knowledge about planaria in the Results section and structuring the writing with summary statements at the end of each section in the Results. It might suggest that authors provide a succinct table with morphological criteria for embryo staging as well as key genes expression changes, accompanied with cartoons.

2) Would it be possible to resolve S2 and S3 by growing the animals at a lower temperature?

3) ISH studies need sense controls. Specifically, in later stages (S6-8), it is unclear whether some of the signal for *piwi1* in Figure 1 could result from probe trapping or non-specific staining in the testes and ovary ducts.

4) Subsection “Progenitors for definitive lineages arise within *piwi*-1+ blastomeres as organogenesis commences” refers to data presumably not shown demonstrating that ~4% of cells expressing PIWI1 by immunostaining at S5 do not harbor *piwi1* transcript, and suggest that the protein perdures after transcript degradation upon differentiation. Can the authors please clarify whether all *piwi*-1+ cells expressed PIWI1 protein in the embryos?

5) My main concern with the authors conclusions lies in the transplant studies. Can the authors please estimate how many cells were transplanted, or how many embryo equivalents. Figure 7 should show a no transplant control (which presumably will not contribute) as well as cNeoblast control from adult (which will presumably contribute more robustly than the embryos). The authors need to demonstrate that *piwi*+ cells and structures rescued following irradiation are donor-derived, as this will dismiss the possibility that the blastomere graft rejuvenates host cells; this can be accomplished genetically by the phenotype of regenerated structures as well as by using a marker such as PKH26 of the donor cells.

Reviewer #3

Davies et al. present a molecular, cellular, and morphological study of planarian (*S. mediterranea*) embryogenesis. They report a molecular resource of stage-enriched genes useful for staging embryos and for studying gene function in embryogenesis, including the discovery of embryo-enriched planarian gene expression. They show that *piwi*+ cells are present throughout embryogenesis, but that lineages for adult tissues only develop within this *piwi*+ population at a particular stage (S5). Interestingly, in transplant experiments, clonogenic (colony-forming) neoblast activity also develops at a particular stage, approximately when lineages appear. The manuscript is an impressive body of work and well written. This paper will be an important landmark in the study of embryogenesis and its comparison to regeneration in planarians.

Abstract:

"Neoblast subpopulations established during embryogenesis persist into adulthood"

Subpopulations may not persist as subpopulations, rather continuously generated. Since the renewal capacity of subpopulations is unknown, different wording is suggested.

Results section:

"seven morphologically distinct stages (S), S2-S8" were selected for sequencing. Panels D-F contain embryos that look superficially similar; thus it would be helpful for the authors to further describe at this point in the text the criteria for stage separation in the sequencing experiment (I assume time of development was also a factor?).

"pairwise comparisons of adjacent time points"

These aren't really "time points", since stages overlap in timing. Different wording is suggested.

Assessment of most statements of data in the text is difficult, because each one requires a goose chase through excel tables, main figures, figure supplements, and online-only figures. Some items called out as figure supplements are not figures but tables with titles different than their figure supplement title. Take subsection “Anarchic, cycling *piwi*-1+ blastomeres fuel Smed embryonic development”, for example; this single sentence refers to 4 panels in 2 main figures, 4 Figure supplements, and 3 supplementary online figures. I tried to find Figure 5—figure supplement 1 to look at the markers for all known and restricted embryonic cell populations and eventually gave up…

Paragraph four in subsection “Anarchic, cycling *piwi*-1+ blastomeres fuel Smed embryonic development”. The paragraph could use a concluding sentence to clarify message.

Similarly, the point of "…sample sizes, the optimal θ' was calculated to be 0.58 +/- 0.33,and was 50-fold more likely to explain the observed trend than a simple normal distribution." is not clear.

Last paragraph of the subsection: restriction of cell cycle activity to *piwi*+ cells in adults should reference the *piwi*-1 2005 paper and could also reference Wagner 2011, which had further analysis of this marker for all dividing cells.

Figure 4—figure supplement 2 Y12 data is interesting, but only S4 staining is apparent. If it is not present in *piwi*+ cells at earlier stages this merits display. The chromatoid body presence could also be considered for a main figure.

Subsection “Early embryo enriched transcripts expressed throughout *piwi*-1+ blastomeres are downregulated as organogenesis begins”. Naming clusters "6" "5" can be deciphered from examination from the figure, but is not immediately clear what these are when reading the text. i.e., individual of the multiple clusters of S2-S4-enriched txts that showed downregulation at S5 and thereafter.

The third paragraph in this section presents analysis of expression of the interesting EEE genes. However, the main messages/synthesis is hard to decipher because of the distracting links out to disparate files/figure locations. The narrative/text feels beholden to a set of files and figures as opposed to the other way around.

More discussion of the EEE genes that are expressed broadly in the embryo would be of interest. What types of genes were discovered?

Reviewer #4

This manuscript contains a large amount of information, much of it contained in supplementary materials. It is an important piece of work, because (1) it addresses the expression of genes on a genome wide scale during the development of a planarian flatworm; (3) it addresses the origin and development of neoblasts.

That being said, in its current form, the manuscript is very difficult to digest. The writing is long (section 5 of the Results alone has 6.5 pages!) and "windy: (see example given for section 2 below), and there are some fundamental questions that are either not addressed, or the answers are "hidden" in the long narrative.

Specific comments:

The Introduction is long and detailed on many aspects of neoblast biology, but at the same time unclear and lacking in other aspects. Generally, I am often not sure whether the authors talk about facts known from previous studies, or are already highlighting the findings to be reported in this paper.

Suggestion: shorten the intro by at least 50%. After (in a briefer format than now) summarizing known functions and types of neoblasts in adults/regeneration, summarize literature (as far as it exists) regarding

-embryogenesis in planarians (see below; the statements about embryogenesis that are now found in section 2 of the Results should be in the Introduction)

-specifically, what is said in the literature where/when do neoblasts first appear in embryos (there will probably not be a lot of published information; one example is Ramachandra et al., 1999, on embryonic BrdU -positive presumed neoblasts in an Acoel).

-proliferation, migration, differentiation of neoblasts during embryonic period.

Results

Section 1: surveys molecular staging; stages themselves described in supplementary table and figures. This is a useful and important work of reference for the planarian developmental field.

Please highlight in the text:

-what in terms of morphogenesis and organogenesis is novel compared to the previous embryological study of *Schmidtea polychroa* (Cardona et al)

-how do stages utilized here (1-8) differ from the 8 previously introduced ones

-what does molecular profiling add to defining stages (clearer temporal distinction?..)

Section 2 – “Anarchic, cycling *piwi*-1+ blastomeres fuel Smed embryonic development”

The section describes the expression pattern of *piwi*, and concludes that the gene is restricted to undifferentiated cells. The text is very difficult to read. Different types of information are intermingled. For example, paragraph two summarizes a result known from the literature. Talks about general expression patterns of non-specified transcripts, documented only in the supplementary material, in differentiating cells. later in the paragraph you speak to the point: *piwi* is expressed in undifferentiated cells in the bodywall at the S3 stage. But how does the pan-nuclear marker EF1a-like 1 show that, since this marker appears to be expressed in all cells (including the *piwi*-positive ones)? The final sentences again talk about findings from the previous literature.

I suggest to remove all references to the previous literature from this (Results) section; put them in the Introduction, and in the Results focus on where exactly *piwi* is expressed. Specifically:

Paragraphs 1-2

-DAPI shows large number of labeled dots: are these all nuclei of blastomeres (Figure 2)?

-if so, only a small subset of blastomeres expresses *piwi* (2A): how would one characterize distribution? Clustered, around embryonic pharynx?

-Paragraph 2: sounds like *piwi* is downregulated in many cells. What is shown (Figure 2) is that from beginning, *piwi* is expressed in only few blastomeres (if DAPI really reflects all nuclei). Is there an earlier stage with many more *piwi*+ cells?

-the pattern and dynamics of *piwi* in undifferentiated cells ("blastomeres") remains a bit vague. Is the pattern the following: stage 1, before embryonic transient cells differentiate: all cells express *piwi*stage 2/3: as embryonic cells differentiate, they lose *piwi*; ALL other cells retain it later stages: as adult cells become postmitotic, they lose *piwi*; ALL other cells retain it

I don't think this is what happens. But the authors do not make it clear in text and figures what exactly happens. I assume that only a small subset of undifferentiated cells express *piwi*. I understand that these are distributed. This information needs to be settled; it is one of the crucial pieces of data required in subsequent sections (see comments below).

Section 3 reports that genes sets known to be associated with adult neoblasts are also expressed in the embryo. This is an important finding. The main question is:

-are these genes, in the embryo (e.g., S4, S5, S6), restricted to *piwi*+ cells?

-are they restricted to ALL undifferentiated cells (assuming that not all undifferentiated cells are *piwi*+; see my questions/comments above)

-are they also expressed in cells that stop dividing and differentiate?

I was not able to find a clear answer to these questions in the text.

Section 4 Early embryonic genes (EEE)

I don't understand the message of this section. One reason for confusion arises right at the beginning:

- in the opening lines the authors talk about "S2-S4 enriched transcripts expressed throughout the *piwi* blastomere population".

- then the authors introduce the 1048 EEE genes: are these the same as above (i.e., they are expressed in the *piwi* blastomere population)?

- because, if they are: why do *piwi* (undifferentiated) blastomeres and differentiated (transient embryonic cells) express the same set of genes? I thought from the previous sections (and on general grounds) that as cells become postmitotic and differentiate, they switch on different genetic programs.

Section 5

It is shown that gene sets associated with specific differentiated cell types are upregulated around stage 5, which is important, but expected, since it confirms the observation that cells differentiate (structurally) at this stage.

The main claim of the heading of this section (that progenitors of specific cell types arise WIHIN the *piwi*-positive population) needs to be demonstrated, and/or described more clearly (see questions to section 2 above)

Section 6

Cell suspensions derived from embryos at different stages (S5-S8) were injected into irradiated adults, and were able to rescue them. This is also an important experiment. A puzzling finding was that cells from S5 embryos were unable to rescue. Is it possible that S5 embryos did not have a high enough density of *piwi*+ cells to do the job? Is it possible to get a value for the number of *piwi*+ cells present in an aliquot of the cell suspension from different stage embryos?

---

## [Author Response]

Reviewer #1

This impressive body of work will be a valuable resource to planarian biologists. To make the manuscript accessible to a broader audience and more manageable for readers in general, here are some editing suggestions that should be addressed before publication:

1) To appeal to the broader eLife audience, the authors should provide more description/context in the main text to help non-experts understand the results presented. This includes data the authors deem not essential for the narrative of the paper and is found in Supplementary Online Material instead of the main text. For example, sphere formation (subsection “Anarchic, cycling piwi-1+ blastomeres fuel Smed embryonic development”) is not defined in the main text, but is described in the SOM as beginning during Stage 2. Furthermore, the distinction between temporary and definitive embryonic organs may not be entirely clear to non-expert readers. The authors do a better job describing this uncommon developmental phenomenon in the SOM (e.g., "The embryonic pharynx degenerates during S6, as the definitive pharynx primordium develops beneath it. Embryonic pharynx markers are no longer detectable by S7").

The manuscript was revised extensively, with the goal of making our story accessible to a broad readership. Care has been taken to eliminate unnecessary specialist vocabulary and to clearly define terms as they are introduced. We shortened the Introduction, eliminating the extensive summary of our results. In addition, we emphasized the foundational resources, key results and conceptual breakthroughs stemming from our work. We added two new paragraphs to the start of the Results section, intended to give readers a crash course in freshwater flatworm embryology. We define necessary terminology and introduce key concepts, like direct development, ectolecithal embryos, dispersed cleavage, temporary embryonic tissues and definitive tissues. We reorganized Figure 2 and Figure 7 (now Figure 2 and Figure 7–Figure 9, respectively) and rewrote the associated text and Figure Legends; we also rewrote much of the text accompanying Figure 5. To aid readers, we reorganized all of the supplemental data so that items are noted chronologically in the text. Supplementary files were broken down into supplemental figures, source data and rich media files. We hope this encourages readers to delve into the wealth of data provided with our manuscript. In addition, we generated an online searchable resource (https://planosphere.stowers.org) that houses the Molecular Staging Series and Molecular Fate Mapping Atlas. This resource, hosted by the Stowers Institute, will be freely available upon publication and will reference our published manuscript.

2) Throughout the manuscript, the authors are urged to present figures in the order they are referred to in the text. Looking up figures (and multiple Supplements) that are "out of order" is cumbersome, especially when considering the sheer volume of data the reader must go through. For example, Figure 1—figure supplement 12 is referred to twice in the text (Results section) before any of the other Figure 1 supplements. Figure 1—figure supplement 12 should be designated as Figure 1—figure supplement 1 (and all other Figure 1 supplements renumbered accordingly thereafter). Also, having two sets of differently named supplemental data (Figure X —figure supplement Y, which is per the journal's style, and Supplementary Online Figure Z, which appears not to be) makes navigating supplementary material unnecessarily difficult for the reader.

To make it easier for readers to navigate through the extensive amount of supplemental data associated with our manuscript, we reorganized and renamed the supplemental data files as suggested by reviewer #1. Figure supplement, source data and rich media files were renamed to reflect the order they are referred to in the text. Large excel tables are now listed as “source data” files; SPIM movies are listed as “rich media” files. In addition, the Molecular Fate Mapping Atlas, formerly the Supplementary Online Material and Supplementary Online Figure 1–Figure 9, are now labeled as [Supplementary-material SD9-data] and Figure 1—figure supplement 11–Figure 1—figure supplement 19. This was done to consolidate the supplementary data, and to eliminate the confusion created by having “figure supplements” and “Supplementary Online Material.”

3) Introduction section: Remove the extensive discussion of results from the Introduction. This will also help shorten the overall length of the paper.

The Introduction was shortened and reorganized, in accordance with suggestions from reviewers #1 and #4. The lines, which contained a detailed summary of our results, were removed from the Introduction.

Reviewer #2

[…]Comments:

1) Overall, the accessibility and readability for a generalist journal as this one should be increased. This can be accomplished by presuming less prior knowledge about planaria in the Results section and structuring the writing with summary statements at the end of each section in the Results. It might suggest that authors provide a succinct table with morphological criteria for embryo staging as well as key genes expression changes, accompanied with cartoons.

We revised the manuscript to make it more accessible to a general audience, and detailed the changes made in this regard in our response to reviewer #1. To facilitate use of this staging system by others, http://planosphere.stowers.org provides cartoons, time (days post-egg capsule deposition), gene expression data (RNA-Seq and WISH), and written descriptions of embryo morphology for each of the stages queried.

2) Would it be possible to resolve S2 and S3 by growing the animals at a lower temperature?

Early S2 embryos undergoing sphere formation are quite fragile and are difficult to detect in live, unstained samples; it’s doubtful that simply lowering the temperature would make these samples easier to find. We are interested in performing gene expression profiling on oocytes, zygotes and early S2 embryos in the future. Laser capture microdissection, performed on fixed tissue stained with a nuclear dye, may be the best route to obtaining tissue for S2 and earlier stages. It is possible that injection of vital dye into oocytes, and/or EdU incorporation into oocytes or early blastomeres, would enable us to identify tissue for RNA-Seq in live samples, and to do lineage tracing studies. We are interested in identifying maternal RNAs contributed to oocytes, the onset of zygotic transcription – which is likely quite early (*i.e.,* before we started our RNA-Seq time course), and determining when the first asymmetries in gene expression occur among the blastomeres. These experiments and are future directions for us.

3) ISH studies need sense controls. Specifically, in later stages (S6-8), it is unclear whether some of the signal for piwi1 in Figure 1 could result from probe trapping or non-specific staining in the testes and ovary ducts.

Hematoxylin and eosin staining is shown in Figure 1, not in situ hybridization data. *piwi-1* expression, shown first in Figure 2, always shows specific expression in parenchymal cells. It is unlikely that the *piwi-1+* signal is nonspecifically trapped in the reproductive system during S6-S8, since the testes, ovaries and somatic gonadal tissue are absent from S8 newborn hatchlings and arise weeks later during juvenile development. Incidentally, we detect *nanos* expression by WISH during S6-S8, in increasing numbers of cells with a noted dorsolateral bias and in the developing eyes (S7-S8). Our suspicion is that the dorsolaterally positioned cells are likely *piwi-1+, nanos+* primordial germ cells; we have not followed up on this point yet, and would need to costain with additional PGC markers as well. Consistent with our WISH results, *nanos* expression was undetectable by RNA-Seq until S5, and was lowly expressed in S5-S8 embryos.

4) Subsection “Progenitors for definitive lineages arise within piwi-1+ blastomeres as organogenesis commences” refers to data presumably not shown demonstrating that ~4% of cells expressing PIWI1 by immunostaining at S5 do not harbor piwi1 transcript, and suggest that the protein perdures after transcript degradation upon differentiation. Can the authors please clarify whether all piwi-1+ cells expressed PIWI1 protein in the embryos?

Yes, *piwi-1+* cells also express PIWI-1 protein (99 +/- 0.4% *piwi-1+* cells were double positive for PIWI-1 protein, n= 4,152 *piwi-1+* cells scored, n= 4 S5 embryos). We amended the text to report this result and created a new figure, Figure 6—figure supplement 1, containing representative images. The quantification you requested is present in the Figure 6—figure supplement 1 legend. The revised main manuscript text is included in our comments to reviewer #1.

5) My main concern with the authors conclusions lies in the transplant studies. Can the authors please estimate how many cells were transplanted, or how many embryo equivalents. Figure 7 should show a no transplant control (which presumably will not contribute) as well as cNeoblast control from adult (which will presumably contribute more robustly than the embryos). The authors need to demonstrate that piwi+ cells and structures rescued following irradiation are donor-derived, as this will dismiss the possibility that the blastomere graft rejuvenates host cells; this can be accomplished genetically by the phenotype of regenerated structures as well as by using a marker such as PKH26 of the donor cells.

We reorganized the presentation of the cell transplantation data and revised the associated text extensively to bring out some of the key points formerly house in the supplemental material. The transplantation experiments are now shown in Figure 7 and 8. In Figure 7, we show quantification for the number of *piwi-1+* cells per host at 1 hour and 5 days post-transplant. We show that comparable numbers of S5, S6, S7 and S8 cells are injected per host; an average injection generally delivers between 50-100 embryonic *piwi-1+* cells into the tail. We detail how the embryonic cell suspensions were prepared in the Materials and methods. We are able to get far fewer S4 embryonic cells injected per host, which was expected since far fewer *piwi-1+* blastomeres are present during S4 versus later stages. The S4 results represent our best efforts to concentrate as many embryo equivalents as possible while minimizing volume of the cell suspension.

We provide scoring data for “no transplant control” data in the text for both sets of transplantation experiments.

We did not run transplantation experiments using adult neoblast enriched fractions as a positive control. We concede that the side by side comparison would be nice to see, though not necessary. There were two reasons for this: 1) We strongly suspected that S8 embryonic cells would rescue since animals at this stage regenerate following amputation (our unpublished observation). The regeneration capacity, coupled with the distribution of *piwi-1+* cells in S8 embryos, suggested strongly to us that upon hatching juvenile worms possess neoblasts. Had we not achieved reproducible rescue results with S8 embryonic donor cells, we certainly would have pursued adult neoblast injection experiments. 2) The published cell transplantation papers used C4 asexual adult animals as donors. Differences in transplanted cell behavior using C4 animals and embryos as donors could arguably be attributed to differences in biotype. While sexually mature adult donor tissue has been shown to rescue in plug graft experiments, we presently don’t have a cell sorting protocol for neoblast isolation from sexually mature adults. Our cell sorting protocols are based on DNA content, as well as cell size and morphology; we would expect our neoblast enriched fraction to be contaminated with dividing germline stem cells and transit amplifying germ cells. This would not be insignificant, given that adults possess roughly 100 testes.

Lethally irradiated adults are capable of wound healing, mount an acute transcriptional response to wounding, and transiently reorganize expression domains of key polarity genes along the A-P axis of the fragment. However, these animals never generate a blastema or regenerate new tissue. None of the lethally irradiated, uninjected sexual adult hosts survived long-term, nor did they mount a regenerative response (e.g., head blastema formation) prior to lysing. The most parsimonious explanation for our results is that engrafted, proliferating, donor-derived *piwi-1+* cells contributed differentiating progeny that subsequently mounted the observed regenerative response in rescue animals. We have seen production of post-mitotic, definitive epidermal progenitors in S5 donor cell transplants (*NB.21.11e+* and *AGAT-1+* cells) at 10 dpt (13 dpi); these definitive epidermal progenitors were present in the tail parenchyma in the neighborhood of *piwi-1+* cells. Moreover, *NB.21.11e+* and *AGAT-1+* cells are irreversibly eliminated by lethal irradiation within 5 dpi, suggesting that the cells present in S5 transplanted hosts at 10 dpt (13 dpi) differentiated from donor-derived *piwi-1+* cells. Additional transplant experiments would be required to examine production of differentiating progeny for other lineages. SNPs could be assessed in donor and transplanted hosts after long-term reconstitution; however, this would require using genetically defined lines for embryonic donor cells, which we did not do for these experiments. We agree that using vital membrane dyes to track donor cells would be a good way of assessing donor versus host contribution following transplantation, and will pursue this suggestion in the future.

Reviewer #3

*[…]Abstract:*

*"Neoblast subpopulations established during embryogenesis persist into adulthood"*

Subpopulations may not persist as subpopulations, rather continuously generated. Since the renewal capacity of subpopulations is unknown, different wording is suggested.

At present we cannot distinguish between the two possibilities mentioned by reviewer #3: self-renewal of committed progenitors within the neoblast population (stable, or persistent subpopulations) or the continual re-emergence of committed progenitors from pluripotent stem cell(s) throughout the lifetime of the animal. The concluding sentences of the Abstract now read:

“Neoblast lineages arise as organogenesis begins and are required for construction of all major organ systems during embryogenesis. These subpopulations are continuously generated during adulthood, where they act as agents of tissue homeostasis and regeneration.”

We also inserted as sentence in to the final paragraph of the Results text associated with Figure 6, which addresses this point:

“At present, we cannot distinguish whether lineages perpetually re-emerge due to asymmetric division of pluripotent stem cells, or whether progenitor populations established during embryogenesis are maintained by self-renewal.”

Results section:

"seven morphologically distinct stages (S), S2-S8" were selected for sequencing. Panels D-F contain embryos that look superficially similar; thus it would be helpful for the authors to further describe at this point in the text the criteria for stage separation in the sequencing experiment (I assume time of development was also a factor?).

Yes, you are correct that a combination of chronology and morphology were used to select samples and define the stages. As stated in the Materials and methods, we rear our animals at 20˚C and collect egg capsules daily. Each collection could theoretically be composed of 0-24 hour old embryos (hours indicating time post egg capsule deposition), but we designate the time of collection as 1 day post-egg capsule deposition and age them out accordingly. The time windows for collection are indicated in Figure 1, and the RNA-Seq replicates ([Supplementary-material SD1-data]) indicate the time point(s) for each of the stages queried.

The main text now reads:

“*Smed* embryos gestate for approximately two weeks at 20˚C prior to hatching. We generated total RNA replicates from single *Smed* embryos for seven chronologically and/or morphologically distinct stages (S), S2-S8 (Figure 1); S1 samples (zygotes, Figure 1—figure supplement 1) were not queried by RNA-Sequencing.”

"pairwise comparisons of adjacent time points"

These aren't really "time points", since stages overlap in timing. Different wording is suggested.

The wording “time points” was changed to “stages.”

Assessment of most statements of data in the text is difficult, because each one requires a goose chase through excel tables, main figures, figure supplements, and online-only figures. Some items called out as figure supplements are not figures but tables with titles different than their figure supplement title. Take subsection “Anarchic, cycling piwi-1+ blastomeres fuel Smed embryonic development”, for example; this single sentence refers to 4 panels in 2 main figures, 4 Figure supplements, and 3 supplementary online figures. I tried to find Figure 5—figure supplement 1 to look at the markers for all known and restricted embryonic cell populations and eventually gave up…

We reorganized and renamed the supplemental data to make it easier for readers to navigate and locate items of interest. Supplemental items are now organized chronologically, as reviewer #1 suggested. All excel data files are now listed as “Figure X—source data X” files. Supplemental figures are listed as “Figure X —figure supplement X.” SPIM movies are now listed as “Rich Media Files.” In addition, the Molecular Fate Mapping Atlas files are now couched under the Figure 1
*subheading, along with the RNA-Seq data. This eliminates a second class of supplemental figure names, which other reviewers also found burdensome. Finally, https://planosphere.stowers.org, an online searchable resource, is another venue for examining much of the supplementary data accompanying our manuscript.*

Paragraph four in subsection “Anarchic, cycling piwi-1+ blastomeres fuel Smed embryonic development”. The paragraph could use a concluding sentence to clarify message.

Similarly, the point of "…sample sizes, the optimal θ' was calculated to be 0.58 +/- 0.33,and was 50-fold more likely to explain the observed trend than a simple normal distribution." is not clear.

Reviewer #1 brought up the same points, and we revised the main text and figure legends to minimize the technical detail and bring forth the main conclusions. The revised text is included earlier in the rebuttal, under the comments to reviewer #1.

Last paragraph of the subsection: restriction of cell cycle activity to piwi+ cells in adults should reference the piwi-1 2005 paper and could also reference Wagner 2011, which had further analysis of this marker for all dividing cells.

We apologize for this oversight and added both references to this passage.

Figure 4—figure supplement 2Y12 data is interesting, but only S4 staining is apparent. If it is not present in piwi+ cells at earlier stages this merits display. The chromatoid body presence could also be considered for a main figure.

Y12 staining was only performed on S4 and S5 embryos. The manuscript was revised as follows:

“Y12 antibodies, which specifically label chromatoid bodies in adult neoblasts {Rouhana, 2012 #482}, stained *piwi-1+* blastomeres during S4-S5 (Figure 4—figure supplement 1, Rich Media File 10).”

Subsection “Early embryo enriched transcripts expressed throughout piwi-1+ blastomeres are downregulated as organogenesis begins”. Naming clusters "6" "5" can be deciphered from examination from the figure, but is not immediately clear what these are when reading the text. i.e., individual of the multiple clusters of S2-S4-enriched txts that showed downregulation at S5 and thereafter.

We altered this paragraph significantly to clarify what we mean by the term “EEE transcripts.” Some of the details regarding expression trends in clusters 5, 6 and 8 were moved the Figure 5 legend.

The main text now reads:

“Hierarchical clustering of S2-S4 enriched transcripts using scaled RPKM values identified 1048 sequences in clusters 5, 6 and 8 that were downregulated by S5 and remained lowly expressed through S8; these sequences are referred to as early embryo enriched (EEE) transcripts (Figure 5, [Supplementary-material SD11-data]). EEE transcripts were likely expressed in blastomeres and/or temporary embryonic tissues, since 98% of the sequences had average expression values at least 5-fold greater in S2 embryos versus Y ([Supplementary-material SD11-data]). Most EEE transcripts were expressed at low levels in intact adults regardless of biotype: average RPKM values less than 1.0 were recorded for 65% and 59% of the EEE transcripts in C4 or SX, respectively ([Supplementary-material SD11-data]).”

The Figure 5 legend now reads:

**“**Hierarchical clustering of S2-S4 enriched transcripts (n=1756) using scaled RPKM data. Left: Heat map. Y: yolk. Colored bars (left) denote clusters 5, 6 and 8 containing early embryo enriched (EEE) transcripts. Cluster 5 sequences (blue, n=413) were expressed at roughly equivalent levels during S2 and S3, with 66% (n=275) transcripts showing 5-fold or greater declines in average expression values between S3 and S5. Cluster 6 sequences (red, n=523) exhibited maximal expression during S2, and average expression levels declined more than 5-fold between S2 and S4 for 81% (n=426) of these transcripts. Cluster 8 sequences (green, n=112) showed peak expression during S4, with 52% (n=60) of the transcripts showing 5-fold or greater declines in average expression values by S5. Right: Normalized expression trends for EEE transcripts in Clusters 5 (blue), 6 (red) and 8 (green) plotted as a function of developmental time. Median 50% of transcripts based on expression maxima are plotted.”

The third paragraph in this section presents analysis of expression of the interesting EEE genes. However, the main messages/synthesis is hard to decipher because of the distracting links out to disparate files/figure locations. The narrative/text feels beholden to a set of files and figures as opposed to the other way around.

We rewrote the section to more clearly articulate our main points. In addition, moved the summary of the WISH expression screen from [Supplementary-material SD11-data] (formerly Figure 5—figure supplement 1) to a separate file, [Supplementary-material SD13-data]. The rewritten text is as follows:

“To determine which cell types express EEE transcripts, and to examine spatiotemporal changes in EEE transcript expression, colorimetric WISH was performed on S2-S8 embryos and intact C4 adults ([Supplementary-material SD13-data]). Some EEE transcripts were expressed exclusively in differentiated temporary embryonic tissues. *VAL-like, MPEG1-like-1, MPEG1-like-2,* and *netrin-like* were solely expressed in the temporary embryonic pharynx until S6, while *gelsolin-like* and *4XLIM-like* were expressed in both the primitive ectoderm and temporary embryonic pharynx during S3-S4 ([Supplementary-material SD13-data], Figure 1—figure supplement 11, Figure 1—figure supplement 12). Expression of these EEE transcripts is likely under zygotic control, occurring during or after the cell fate decisions to downregulate *piwi-1,* exit the cell cycle and differentiate.

Most EEE transcripts queried by WISH (n=15, 75% assayed) were expressed in both undifferentiated blastomeres and temporary embryonic tissue(s) during S3-S4 (Figure 5, [Supplementary-material SD13-data]). Some of these transcripts may be maternally deposited, albeit we cannot ascertain the relative contribution(s) of maternal and zygotic expression from our RNA-Sequencing data. Expression of these EEE transcripts diminished greatly by S5, with moderate and lowly expressed EEE transcripts becoming undetectable; specific expression of robustly expressed transcripts sometimes persisted until S6 (Figure 5, [Supplementary-material SD13-data]). Consistent with the RNA-Seq and Nanostring nCounter results, EEE transcript expression was not detected by colorimetric WISH in S7, S8 embryos or C4 adults (Figure 5, [Supplementary-material SD13-data]). Fluorescent double WISH performed with riboprobes complementary to *piwi-1* and the EEE transcripts *tct-like, BTF3-like, DDX5-like* and *eIF4a-like* revealed coincident expression throughout the S4 blastomere compartment (Figure 5). Intriguingly, EEE transcript expression often decayed quicker in differentiated cells than undifferentiated blastomeres, raising the possibility that regulation of EEE transcription and/or transcript stability may vary by cell type. While robust expression of many EEE transcripts was detected in blastomeres during S3-S4, EEE transcript expression in temporary embryonic tissues was present during S3 and drastically diminished by S4 (Figure 5, [Supplementary-material SD13-data]).

EEE transcripts expressed throughout the undifferentiated *piwi-1+* blastomere population in S3-S4 embryos are downregulated as definitive organogenesis begins during S5. These transcripts likely represent a key temporal shift in the expression profile of *piwi-1+* cells during embryogenesis. Moreover, EEE transcript expression provides a molecular metric to distinguish *piwi-1+* blastomeres from adult neoblasts.”

More discussion of the EEE genes that are expressed broadly in the embryo would be of interest. What types of genes were discovered?

We have validated expression of a relatively modest cohort of EEE transcripts by WISH, and will do more extensive work to rigorously identify homologs, validate expression of additional candidates and ultimately will devise RNAi knock-down methodology to query function of these genes during S2-S4. The EEE transcripts that were specifically expressed in temporary embryonic tissues clustered with those that showed pan-embryonic expression during S3, followed by blastomere-specific expression during S4; that may confound attempts to hypothesize much based on the annotations alone. Roughly a third of the EEE transcripts have no BLASTx hit versus the NR or Swiss-Prot databases, and are considered novel; we have done little to pursue these candidates. Of the candidates that have BLASTx-based annotations, many appear to be involved in translational control and proteostasis; these candidates superficially appear similar to the types of maternally deposited transcripts present in other organisms, but of course we cannot confirm or refute that connection at this point. The annotated lists are provided in [Supplementary-material SD11-data].

Reviewer #4

[…]Specific comments:

The Introduction is long and detailed on many aspects of neoblast biology, but at the same time unclear and lacking in other aspects. Generally, I am often not sure whether the authors talk about facts known from previous studies, or are already highlighting the findings to be reported in this paper.

*Suggestion: shorten the intro by at least 50%. After (in a briefer format than now) summarizing known functions and types of neoblasts in adults/regeneration, summarize literature (as far as it exists) regarding*

*-embryogenesis in planarians (see below; the statements about embryogenesis that are now found in section 2 of the Results should be in the Introduction)*

-specifically, what is said in the literature where/when do neoblasts first appear in embryos (there will probably not be a lot of published information; one example is Ramachandra et al., 1999, on embryonic BrdU -positive presumed neoblasts in an Acoel).

-proliferation, migration, differentiation of neoblasts during embryonic period.

The Introduction was shortened in accordance with comments from reviewers #1, 2 and 4. The lengthy discussion of our results was removed from the Introduction. Instead, we highlight our main conclusions and conceptual advances at the end of the revised Introduction. Information pertaining to neoblasts – molecular and functional definitions, functions, subclasses, and roles during adulthood – were condensed and reorganized, as suggested by reviewer #4.

In the interest of making the manuscript more accessible to a general audience, we added two new paragraphs to the start of the Results section which introduce some of the essential concepts and terminology for Triclad embryos (e.g., direct developer, ectolecithal embryo, dispersed cleavage, temporary embryonic tissues, definitive organs).

Results

Section 1: surveys molecular staging; stages themselves described in supplementary table and figures. This is a useful and important work of reference for the planarian developmental field.

Please highlight in the text:

*-what in terms of morphogenesis and organogenesis is novel compared to the previous embryological study of Schmidtea polychroa (Cardona et al)*

*-how do stages utilized here (1-8) differ from the 8 previously introduced ones*

We are happy to comment on similarities and distinctions between our staging series and the published staging series for *S. polychroa (Spol).* However, we do not expound on these points in the Results section because they will likely appeal to specialists and not the broader *eLife* readership. Note that it is difficult to directly compare developmental timing with those of the 8 stages presented in the Cardona et al. 2005 publication because the *Spol* animals were reared at 17˚C and had a gestation time of around 20 days. To confound matters further, the Romero lab later published another *Spol* staging series for animals reared at 20˚C (Martin-Duran et al. 2010), in which only 7 stages are articulated.

*Smed* Stage 1/*Spol* Stage 1 (Cardona et al. 2005)/Not present in (Martin-Duran et al. 2010): We do not have a molecular signature for Stage1 and we have not looked at the zygote or first cell divisions in detail. Commenting further at this point is difficult. In addition, *Spol* animals are parthenogenic while *Smed* reproduce sexually, so gametogenesis, egg activation and zygote formation likely differ between the species.

*Smed* Stage 2/*Spol* Stage 2 (Cardona et al. 2005)/Stage 1(Martin-Duran et al. 2010): Both *Smed* and *Spol* embryos are ectolecithal developers, have blastomeres that undergo dispersed cleavage and form spheres. We believe asymmetries are present within the blastomere population during S2, but disagree with the “Type 1” and “Type 2” blastomere designations presented in Cardona et al. 2005, and which were eventually disavowed in later *Spol* publications. We cannot comment on the molecular mechanisms implicated in sphere formation at present.

*Smed* Stages 3+4/*Spol* Stage 3(Cardona et al. 2005)/ *Spol* Stage 2+3 (Martin-Duran et al. 2010): Our molecular profiling data supports our designation for two separate stages. *Smed* Stage 3 is likely most analogous to *Spol* Stage 2 (Martin-Duran et al. 2010), while *Smed* Stage 4 is likely most analogous to *Spol* Stage 3 (Martin-Duran et al. 2010).

*Smed* Stage 5/*Spol* Stage 4-5(Cardona et al. 2005)/*Spol* Stage 4 (Martin-Duran et al. 2010): Gut development is clearly underway during Stage 5, but since these gastrodermal cells express novel embryonic markers it is difficult to say without further analysis whether this is a temporary or definitive tissue. Mature gut progenitors arise at this stage, but the gastrodermal cells lining the intestinal cavity do not express these markers. While we agree that more blastomeres, and hence more mitotic cells are present during S5 than earlier stages, our analysis of the mitotic index for the blastomere compartment suggests that the S3-S5 *Smed* blastomeres divide at a constant rate, arguing gainst the “mitotic burst” phenomenon described in *Spol* references.

*Smed* Stage 6-8/*Spol* Stages 6-8 (Cardona et al. 2005): these stages are superficially similar, though distinctions in timing and morphology may be present for different definitive organ systems mentioned.

*Smed* Stage 6-7/Stage 5 (Martin-Duran et al. 2010): Elongation may occur, on average, much sooner in *Spol* embryos than *Smed* embryos.

*Smed* Stage 7-8/Stage 7 (Martin-Duran et al. 2010): Definitive organ development and maturation may occur, on average, much sooner in *Spol* embryos than *Smed* embryos.

It would certainly be possible to conduct similar expression profiling studies in *Spol* and/or other freshwater planarian embryos, and/or to examine spatiotemporal changes in expression patterns during embryogenesis for homologs of the markers we present in this publication. Comparative analysis of gene function during embryogenesis would also be required for informed comment on many of these issues.

-what does molecular profiling add to defining stages (clearer temporal distinction?..)

The molecular profiling data suggests that the genetic regulatory networks and molecular mechanisms governing *Smed* embryogenesis are robust, and likely temporally much more precise than was anticipated from previous studies of freshwater flatworm embryogenesis. Our RNA-Sequencing data underscores the profound developmental transition between Stages 4 and 5, one that is not readily apparent by changes in morphology alone. We were also able to identify new classes of embryo enriched transcripts which were not anticipated from previous studies. Moreover, the gene expression signatures for each stage provide a wealth of candidates for novel tissue and/or stage specific markers and putative genetic regulators whose function(s) will be assessed in future studies.

*Section 2 – “Anarchic, cycling piwi-1+ blastomeres fuel Smed embryonic development”*

The section describes the expression pattern of piwi, and concludes that the gene is restricted to undifferentiated cells. The text is very difficult to read. Different types of information are intermingled. For example, paragraph two summarizes a result known from the literature. Talks about general expression patterns of non-specified transcripts, documented only in the supplementary material, in differentiating cells. later in the paragraph you speak to the point: piwi is expressed in undifferentiated cells in the bodywall at the S3 stage. But how does the pan-nuclear marker EF1a-like 1 show that, since this marker appears to be expressed in all cells (including the piwi-positive ones)?

The data presented in Figure 2 was reorganized with *EF1a-like-1* expression depicted in Figure 2, and *piwi-1* expression depicted thereafter. The text was revised to clarify some of the important points brought up by reviewers #1 and #4. The first paragraph of the section, “An anarchic, cycling *piwi-1+* blastomeres fuel *Smed* embryonic development,” describes the *EF1a-like-1* expression in S2 and S3 embryos, and includes detailed descriptions of embryonic cell types and architecture. *EF1a-like-1* is a pan-embryonic marker, meaning that it is expressed in differentiated temporary embryonic tissues (the primitive ectoderm, the temporary embryonic pharynx, the primitive gut) and the undifferentiated blastomeres located in the embryonic wall (the parenchymal space between the primitive ectoderm and endoderm) of S3 embryos. *EF1a-like-1* is not expressed in yolk cells. Yolk cells are present in the embryonic wall along with the blastomeres; these cells have very small, condensed sytox green+ nuclei. Ingested yolk is also present in the primitive gut cavity. Blastomere nuclei are larger, with diffuse sytox green signal (likely due to the decondensed nature of the chromatin), are easily distinguished from yolk cell nuclei in the embryonic wall. In addition, the blastomere nuclei are readily distinguished from primitive ectodermal nuclei by sytox green staining; latter are stretched and sit at the surface of the sphere. We are confident that the combination of *EF1a-like-1* and nuclear staining makes allows for unambiguous identification of undifferentiated blastomeres in the embryonic wall of S3 embryos. *Piwi-1* expression was always restricted to, and expressed throughout, the undifferentiated blastomere compartment of S3 embryos in *EF1a-like-1, piwi-1* costaining experiments. Quantification is provided in the Figure 2 legend.

The final sentences again talk about findings from the previous literature.

I suggest to remove all references to the previous literature from this (Results) section; put them in the Introduction, and in the Results focus on where exactly piwi is expressed. Specifically:

Paragraphs 1-2

-DAPI shows large number of labeled dots: are these all nuclei of blastomeres (Figure 2)?

-if so, only a small subset of blastomeres expresses piwi (2A): how would one characterize distribution? Clustered, around embryonic pharynx?

No, all of the nuclei are NOT blastomeres in the former Figure 2 (now 2D and 2B, respectively). Yolk cells are abundant but do not express either *EF1a-like-1* or *piwi-1.* Please refer to the previous paragraph regarding the *EF1a-like-1,* sytox green costaining results.

The distribution of blastomeres in S3-S4 embryos varies widely, as described in Figure 2. At present we cannot comment on the distribution of the blastomeres during S2 as the embryos undergo sphere formation; that would require costaining with additional markers specific for the temporary embryonic tissues.

-Paragraph 2: sounds like piwi is downregulated in many cells. What is shown (Figure 2) is that from beginning, piwi is expressed in only few blastomeres (if DAPI really reflects all nuclei). Is there an earlier stage with many more piwi+ cells?

*-the pattern and dynamics of piwi in undifferentiated cells ("blastomeres") remains a bit vague. Is the pattern the following: stage 1, before embryonic transient cells differentiate: all cells express piwistage 2/3: as embryonic cells differentiate, they lose piwi; ALL other cells retain itlater stages: as adult cells become postmitotic, they lose piwi; ALL other cells retain it*

I don't think this is what happens. But the authors do not make it clear in text and figures what exactly happens. I assume that only a small subset of undifferentiated cells express piwi. I understand that these are distributed. This information needs to be settled; it is one of the crucial pieces of data required in subsequent sections (see comments below).

We show that *piwi-1* mRNA is maternally deposited in oocytes (Figure 2, revised manuscript), and that *piwi-1+* blastomeres undergo dispersed cleavage during S2, prior to sphere formation (Figure 2, revised manuscript). As noted previously, nuclear dyes mark both blastomere and yolk cell nuclei. Our data suggests that *piwi-1* is expressed in the zygote and its descendants, the blastomeres, during S2. During sphere formation, asymmetries in the *piwi-1+* blastomere population become evident: some blastomeres differentiate into temporary embryonic tissues, and these cells down-regulate *piwi-1* and exit the cell cycle. We do not know when asymmetries within the *piwi-1+* blastomeres develop during S2 and would like to explore this in the future. However, we never detect expression of *piwi-1* (Figure 2, revised manuscript) or the cell cycle markers *PCNA* or *RNRM2-2* (Figure 3, revised manuscript) in differentiated tissues of S2 or S3 embryos (compare to the expression pattern for *EF1a-like-1* in Figure 2, revised manuscript). During S3 and thereafter, *piwi-1* expression is restricted to undifferentiated cells. We agree with reviewer #4 that clear communication of these points is critical, and revised the text associated with Figure 2 (revised manuscript) as follows:

*“piwi-1+* cells were present throughout embryogenesis. *piwi-1* expression was detected in oocytes (Figure 2), suggesting that zygotes contain *piwi-1* mRNA. Zygote-derived blastomeres undergoing dispersed cleavage among yolk cells during S2 also expressed *piwi-1* (Figure 2). Costaining with *piwi-1* riboprobes and antibodies raised against the G2-M phase mitotic marker H3S10p showed that *piwi-1+* blastomeres divide asynchronously during S2 (Figure 2).

As spheres form during S2, some blastomeres downregulate *piwi-1* expression and differentiate into temporary embryonic cell types. *piwi-1* expression was restricted to, and expressed throughout, the undifferentiated blastomere population in S3 embryos, as demonstrated by double fluorescent WISH with *EF1a-like-1* and *piwi-1* riboprobes(Figure 2, Rich Media File 2). *piwi-1+* blastomeres were always located in the embryonic wall, sandwiched between the primitive ectoderm and endoderm (Figure 2). *piwi-1* expression was never detected in primitive ectodermal cells, the embryonic pharynx or primitive gut cells (Figure 2). During S3-S5, *piwi-1+* cell number clearly increased, effectively blanketing the sphere (Figure 2). As definitive gut development proceeded during S6-S8, *piwi-1+* cells occupied the parenchyma between the developing gut branches (Figure 2). Notably, *piwi-1+* cells were not detected in the definitive pharynx, and the compartment receded from the anterior margin as head structures developed (Figure 2). By S8, the spatial distribution of *piwi-1+* cells was indistinguishable from that of the adult neoblast compartment (Figure 2).”

*Section 3 reports that genes sets known to be associated with adult neoblasts are also expressed in the embryo. This is an important finding. The main question is:*

*-are these genes, in the embryo (e.g., S4, S5, S6), restricted to piwi+ cells?*

*-are they restricted to ALL undifferentiated cells (assuming that not all undifferentiated cells are piwi+; see my questions/comments above)*

*-are they also expressed in cells that stop dividing and differentiate?*

I was not able to find a clear answer to these questions in the text.

We show colocalization analysis for *piwi-1* and several adult asexual neoblast-enriched genes, including *piwi-2, piwi-3, bruli-1* and *tud-1* in S3-S5 embryos (Figure 4). The staining patterns are coincident at these stages, meaning that all *piwi-1+* cells express these markers (and vice versa). These data suggest that all undifferentiated cells express a similar complement of nuage genes in S3-S5 embryos: they are pan-blastomere markers at early stages, just as they are expressed throughout the neoblast population in adulthood. The numbers in the lower right corner of the merged images display the percent colocalization with respect to both populations scored (see Figure 4 legend); very few single positive cells were observed. We have not performed colocalization analysis and quantification on S6 and older embryos, but would predict to see virtually all *piwi-1+* cells at later stages expressing these markers, as do adult neoblasts.

Some differentiating and/or terminally differentiated cells may retain expression of some of these markers. For example, *bruli-1* expression was detected in the four neurons innervating the temporary embryonic pharynx (Figure 4), and may be expressed in the brain and potentially other neural tissues in late stage embryos, as in adults. Similarly, *piwi-2* expression was frequently seen in the S3 temporary embryonic pharynx (Figure 4), and may also be expressed in neural tissues in late stage embryos, as seen in adults. In the future, RNA-Seq on cycling cells versus differentiated cells or whole embryos (WT or irradiated) may yield a global view of how enriched or exclusive expression of these genes are to the blastomere/neoblast compartment.

Section 4 Early embryonic genes (EEE)

I don't understand the message of this section. One reason for confusion arises right at the beginning:

- in the opening lines the authors talk about "S2-S4 enriched transcripts expressed throughout the piwi blastomere population".

- then the authors introduce the 1048 EEE genes: are these the same as above (i.e., they are expressed in the piwi blastomere population)?

The 1048 EEE transcripts are a subset of the 1756 S2-S4 enriched transcripts that were clustered in Figure 5. Specifically, these 1048 transcripts comprise clusters 5, 6 and 8. Details regarding the expression profiles and BLASTx-based annotation of the EEE transcripts are found in [Supplementary-material SD11-data] (formerly Figure 5—figure supplement 1).

For clarification, we rewrote the first paragraph of this section and moved some of the detailed description of the cluster 5, 6 and 8 expression trends to the Figure 5 legend. The revised text for both the manuscript and figure legend are detailed in a response to reviewer #3, who also brought up this point.

- because, if they are: why do piwi (undifferentiated) blastomeres and differentiated (transient embryonic cells) express the same set of genes? I thought from the previous sections (and on general grounds) that as cells become postmitotic and differentiate, they switch on different genetic programs.

In response to comments from reviewer #3, we rewrote the section detailing the WISH screen and costaining experiments described in Figure 5; please see the previous section for the revised text. Maternal deposition of some EEE transcripts may explain why undifferentiated blastomeres and differentiated temporary embryonic cell types in S3 embryos co-express some EEE transcripts. However, we note that expression of these EEE transcript diminishes quicker in the temporary embryonic tissues than in the blastomeres. Differential transcript stability, relative contributions of maternal load and zygotic transcription, and different modes of maternal mRNA degradation may all be at play and could be investigated in the future. We also note that some EEE transcripts do appear to be expressed exclusively in differentiated cell types, like the temporary embryonic pharynx and the primitive ectoderm. This is consistent with the idea that cells turn on different genetic programs as they differentiate, and suggests that these EEE transcripts are transcribed zygotically, during or after the cell fate decision to downreguate piwi-1, exit the cell cycle and differentiate.

Section 5

It is shown that gene sets associated with specific differentiated cell types are upregulated around stage 5, which is important, but expected, since it confirms the observation that cells differentiate (structurally) at this stage.

The main claim of the heading of this section (that progenitors of specific cell types arise WIHIN the piwi-positive population) needs to be demonstrated, and/or described more clearly (see questions to section 2 above)

Great care was taken to articulate each point critical to our model of how the diversity of lineages required for organogenesis are born during *Smed* embryogenesis. Moreover, our results immediately suggest why gastrulation is dispensible for freshwater planarian embryos. While many details remain to be worked out in the future, we are energized by paradigm-challenging ramifications of our work. We made very modest changes to the text for this section, as we contend that reviewer #4’s difficulty understanding the conclusions were rooted in not in our writing, but her/his confusion about ectolecithal development and whether *piwi-1* was expressed throughout the undifferented blastomere population (see comments related to Figure 2).

Section 6

*Cell suspensions derived from embryos at different stages (S5-S8) were injected into irradiated adults, and were able to rescue them. This is also an important experiment. A puzzling finding was that cells from S5 embryos were unable to rescue. Is it possible that S5 embryos did not have a high enough density of piwi+ cells to do the job? Is it possible to get a value for the number of piwi+ cells present in an aliquot of the cell suspension from different stage embryos?*

We show that comparable numbers of *piwi-1+* cells were introduced per host for S5 and later stages (see Revised Figure 7, quantification for the number of *piwi-1+* cells per host at 1 hour post-transplant). This data was present in the supplemental material in our initial submission, and was moved to a main figure due to its importance for our conclusion regarding stage-dependent differences in donor cell activity.